# Protein Phosphatase 1 inactivates Mps1 to ensure efficient Spindle Assembly Checkpoint silencing

**Margarida Moura[1,2†], Mariana Osswald[1,2†], Nelson Leça[1,2], João Barbosa[1,2], António J Pereira[1,2], Helder Maiato[1,2,3], Claudio E Sunkel[1,2,4*], Carlos Conde[1,2*]**

[1]i3S, Instituto de Investigação e Inovação em Saúde, Universidade do Porto, Porto, Portugal; [2]Instituto de Biologia Molecular e Celular, Universidade do Porto, Porto, Portugal; [3]Departamento de Biomedicina, Unidade de Biologia Experimental, FMUP - Faculdade de Medicina da Universidade do Porto, Porto, Portugal; [4]Departamento de Biologia Molecular, ICBAS - Instituto de Ciências Biomédicas Abel Salazar, Porto, Portugal

**Abstract** Faithfull genome partitioning during cell division relies on the Spindle Assembly Checkpoint (SAC), a conserved signaling pathway that delays anaphase onset until all chromosomes are attached to spindle microtubules. Mps1 kinase is an upstream SAC regulator that promotes the assembly of an anaphase inhibitor through a sequential multi-target phosphorylation cascade. Thus, the SAC is highly responsive to Mps1, whose activity peaks in early mitosis as a result of its T-loop autophosphorylation. However, the mechanism controlling Mps1 inactivation once kinetochores attach to microtubules and the SAC is satisfied remains unknown. Here we show *in vitro* and in *Drosophila* that Protein Phosphatase 1 (PP1) inactivates Mps1 by dephosphorylating its T-loop. PP1-mediated dephosphorylation of Mps1 occurs at kinetochores and in the cytosol, and inactivation of both pools of Mps1 during metaphase is essential to ensure prompt and efficient SAC silencing. Overall, our findings uncover a mechanism of SAC inactivation required for timely mitotic exit.

**\*For correspondence:** cesunkel@
ibmc.up.pt (CES); cconde@ibmc.
up.pt (CC)

[†]These authors contributed
equally to this work

**Competing interests:** The
authors declare that no
competing interests exist.

**Reviewing editor:** Andrea
Musacchio, Max Planck Institute
of Molecular Physiology,
Germany

## Introduction

Accurate chromosome segregation depends on the Spindle Assembly Checkpoint (SAC), a conserved signaling pathway that monitors the attachment of kinetochores to microtubules and prevents sister chromatid separation until all chromosomes are attached to microtubules emanating from opposite spindle poles (*Musacchio and Salmon, 2007*; *Foley and Kapoor, 2013*). Mps1 kinase is a master regulator of SAC function (*Liu and Winey, 2012*). Mps1 phosphorylates MELT motifs on Spc105/KNL1 creating docking sites for the hierarchical recruitment of additional SAC proteins required for the assembly of the Mitotic Checkpoint Complex (MCC), a tetrameric protein complex that inhibits the Anaphase Promoting Complex (APC/C) and thereby prevents anaphase onset (*Shepperd et al., 2012*; *Primorac et al., 2013*; *Vleugel et al., 2015*; *London et al., 2012*; *Yamagishi et al., 2012*). Moreover, Mps1-mediated phosphorylation of Mad1 dramatically accelerates the structural conversion of Mad2 into its closed conformer, the rate-limiting step for MCC assembly and thus, of SAC activation (*Faesen et al., 2017*; *Ji et al., 2017*). The recruitment of Mps1 to unattached kinetochores is directly mediated by the Ndc80 complex and promoted by Aurora B kinase (*Santaguida et al., 2011*; *Saurin et al., 2011*; *Conde et al., 2013*; *Nijenhuis et al., 2013*). This localized accumulation of Mps1 is thought to potentiate its trans-autophosphorylation on the

T-loop to ensure full kinase activity and robust SAC signaling (*Kang et al., 2007*; *Sacristan and Kops, 2015*).

The events that underlie SAC inactivation are less well understood. The phosphatases PP1-γ and PP2A-B56 appear to play an important role in this process by dephosphorylating KNL1-MELT motifs (*London et al., 2012*; *Espert et al., 2014*; *Nijenhuis et al., 2014*), which contributes to the removal of SAC proteins from kinetochores following microtubule attachments (*Etemad and Kops, 2016*). Furthermore, elegant *in vitro* studies demonstrated that binding of Mps1 to the calponin homology (CH) domains of Ndc80 and Nuf2 is inhibited by microtubules (*Hiruma et al., 2015*; *Ji et al., 2015*). Although this competition mechanism precludes the recruitment of Mps1 to bioriented kineto-chores, it is however insufficient to remove all Mps1 from kinetochores even after the formation of stable end-on attachments (*Aravamudhan et al., 2015*; *Hiruma et al., 2015*; *Ji et al., 2015*). Work in budding yeast proposes that end-on attachments physically separate residual Mps1 from the Spc105/KNL1 phosphodomain, hence disrupting MELT phosphorylation at metaphase kinetochores (*Aravamudhan et al., 2015*). While this may contribute to silence the SAC in yeast, this model does not account for the highly dynamic behaviour of active Mps1 that has been observed in human cells (*Jelluma et al., 2010*). The association of Mps1 with the Ndc80 complex is very transient and the kinase diffuses rapidly into the cytoplasm, which is likely to allow Mps1 to overcome the increased spatial separation between its kinetochore receptor and Spc105/KNL1 MELT motifs. Moreover, kinetochore recruitment of Mad1-Mad2 in metazoa is also mediated through an Spc105/KNL1-inde-pendent pathway (*Schittenhelm et al., 2009*; *Caldas et al., 2015*; *Silió et al., 2015*). This is known to rely on the recruitment of the Rod/ZW10/Zwilch (RZZ) complex to unattached kinetochores, an event that is also controlled by Mps1 kinase (*Santaguida et al., 2010*). Importantly, in addition to its role at unattached kinetochores, Mps1 also contributes to SAC signaling in the cytoplasm. Soluble Mps1 is required for the assembly of pre-mitotic MCC that determines the peak time of anaphase onset in the absence of kinetochore–microtubule attachment problems (*Rodriguez-Bravo et al., 2014*). Moreover, Mps1 lacking its kinetochore-binding domain is sufficient to delay mitotic exit in mouse embryonic fibroblasts and RPE1 cells challenged with spindle poisons (*Foijer et al., 2014*; *Maciejowski et al., 2010*) and cytosolic Mps1 was shown to be required to support SAC arrest caused by kinetochore-tethered Mad1 (*Maldonado and Kapoor, 2011*). Therefore, in addition to the control exerted on Mps1 kinetochore recruitment and to alterations on kinetochore architecture, other mechanisms must contribute to disrupt SAC signaling upon stable end-on microtubule attach-ments. Here we show that in *Drosophila*, PP1-87B/PP1-γ dephosphorylates the T-loop of cytosolic and kinetochore-associated Mps1. This renders both pools of Mps1 inactive during metaphase, which we find to be critical for rapid SAC silencing and timely mitotic exit.

## Results

### Depletion of PP1-87B results in a metaphase delay with stable kinetochore-microtubule attachments

In yeast and human cells, KNL1-bound PP1-γ antagonizes Mps1 signaling at kinetochores through dephosphorylation of MELT motifs, which leads to a reduction in Bub and Mad proteins at meta-phase kinetochores (*London et al., 2012*; *Zhang et al., 2014*; *Nijenhuis et al., 2014*). Since *Drosophila* Spc105/KNL1 inherently lacks phospho-regulatable MELTs (*Schittenhelm et al., 2009*; *Conde et al., 2013*) but PP1 is still required for timely mitotic exit in flies (*Chen et al., 2007*), we resorted to this model organism to uncover novel SAC-silencing mechanisms that might have been overlooked in other systems.

Live cell-imaging analysis showed that depletion of PP1-87B orthologue from *Drosophila* S2 cells results in an abnormal accumulation of Mad1 at aligned kinetochores and a pronounced extension of the metaphase duration (*Figure 1A,B*; *Figure 1—figure supplement 1* and *Videos 1* and *2*). Co-depletion of Mps1 prevented the anaphase onset delay, indicating it is due to active SAC signaling (*Figure 1A,B*; *Figure 1—figure supplement 1* and *Video 3*). Aurora B phosphorylates KMN proteins to destabilize erroneous kinetochore-microtubules interactions (*Liu et al., 2009*; *Welburn et al., 2010*; *Lampson and Cheeseman, 2011*). Since in human cells PP1 was shown to modulate Aurora B activity by antagonizing its activating T232 autophosphorylation (*Liu et al., 2010*; *Posch et al., 2010*; *Wurzenberger et al., 2012*), the metaphase delay observed in PP1-87B depleted cells could

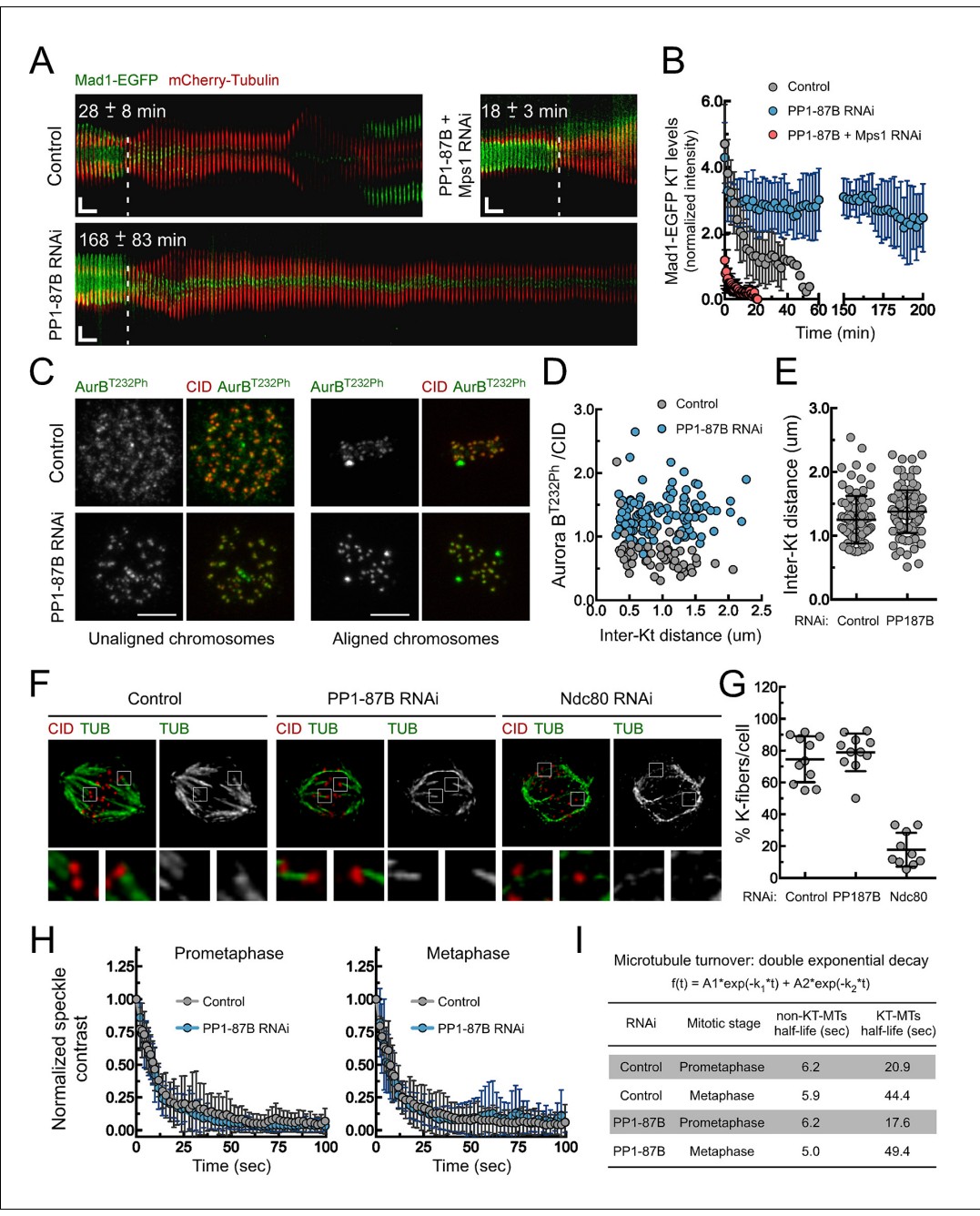

**Figure 1.** Depletion of PP1-87B results in a SAC-dependent metaphase delay despite stable kinetochore-microtubule attachments. (**A,B**) Kymograph analysis of mitotic progression monitored by live-cell imaging (**A**) and corresponding quantification of Mad1-EGFP kinetochore levels (**B**) of control, PP1-87B depleted or Mps1 and PP1-87B co-depleted *Drosophila* S2 cells expressing mCherry-Tubulin and Mad1-EGFP under control of *Mad1* promoter. The mean time from nuclear envelope breakdown to anaphase onset for each condition is displayed in the corresponding kymograph. Mad1-EGFP fluorescence intensities at kinetochores were corrected for cytosolic signal (N ≥ 9 cells for each condition). Vertical dashed line indicates the frame corresponding to nuclear envelope breakdown. Horizontal scale bar: 5 min. Vertical scale bar: 5 μm. (**C,D**) Representative immunofluorescence images (**C**) and corresponding quantifications (**D**) of inter-kinetochore distances and relative levels of Aurora B T232 phosphorylation (AurB$^{T232Ph}$) at unaligned and aligned chromosomes in control and PP1-87B depleted S2 cells. AurB$^{T232Ph}$ relative levels were plotted over the inter-kinetochore distance measured as the distance between centroids of CID pairs. AurB$^{T232Ph}$ fluorescence intensities were determined relative to CID signal (N ≥ 58 kinetochore pairs for each condition). Scale bar: 5 μm. (**E**) Quantification of inter-kinetochore distances measured in chromosomes aligned at the metaphase plate of control and PP1-87B depleted S2 cells. Inter-kinetochore

*Figure 1 continued on next page*

*Figure 1 continued*

distances were measured as the distance between centroids of identified CID pairs (N $\geq$ 86 kinetochores for each condition). (F,G) Representative immunofluorescence images (F) and corresponding quantification (G) of cold-stable kinetochore fibers in control, PP1-87B and Ndc80 depleted S2 cells. CID immunolocalization was used as kinetochore reference. The insets display magnifications of the outlined regions. The graph represents the % of kinetochores attached to cold-stable microtubules per cell (N $\geq$ 10 cells for each condition). (H,I) Analysis of microtubules turnover rates by speckle contrast fadeout of GFP-$\alpha$-Tubulin. (H) Contrast fadeout–time curves of GFP-$\alpha$-Tubulin fluorescent speckles (lines) and their time point means (dots) measured in rectangular areas enclosing the spindle of control and PP1-87B-depleted S2 cells in prometaphase and metaphase. The rate of speckle contrast fadeout was calculated to obtain microtubule turnover rates. (I) Table showing microtubules half-lives of non-kinetochore- (non-KT-MTs) and kinetochore-microtubules (KT-MTs) of control and PP1-87B-depleted S2 cells determined by inducible speckle imaging in prometaphase and metaphase. The average speckle intensity squared-contrast at each time point was fit to a double-exponential curve A1*exp(-k1*t)+A2*exp(-k2*t), in which t is time, A1 represent the less stable population (non-KT-MTs) and A2 the more stable population (KT-MTs) with decay rates of k1 and k2, respectively. The turnover half-life for each population was calculated as ln2/k (N $\geq$ 7 metaphase cells for each condition). Data information: in (A), (B), (E), (G) and (H) data are presented as mean ± SD. Numerical source data for this figure are provided in *Figure 1—source data 1*.

The following source data and figure supplements are available for figure 1:

**Source data 1.** Numerical data for *Figure 1*.
**Figure supplement 1.** Mad1-EGFP localization pattern during mitosis in S2 cells depleted of PP1-87B.
**Figure supplement 2.** Analysis of microtubule turnover in control and PP1-87B depleted metaphase cells by speckle contrast fade-out.

be caused by labile kinetochore-microtubule attachments continuously engaging the SAC. We found that depletion of PP1-87B from *Drosophila* S2 cells increased Aurora B activation at centromeres/kinetochores of prometaphase and metaphase chromosomes, as expected (*Figure 1C,D*). However, the inter-kinetochore distances on aligned chromosomes were similar to those measured in control cells (*Figure 1E*), indicating that kinetochores are under mechanical tension compatible with stably attached microtubules (*Li and Nicklas, 1995*; *Akiyoshi et al., 2010*; *Khodjakov and Pines, 2010*; *Ye et al., 2016*). In agreement, PP1-87B depleted cells revealed the presence of cold stable kinetochore-fibers, as observed in control cells (*Figure 1F,G*). Furthermore, we quantified the stability of kinetochore-microtubule attachments by measuring microtubule turnover rates with inducible speckle imaging (ISI) (*Pereira et al., 2016*). Briefly, a speckled laser beam imprints a bleaching pattern on GFP-$\alpha$-Tubulin that dissipates (by fade-out of the bright spots and fade-in of the dark spots) in a timescale indicative of microtubule lifetime. In contrast to conventional photobleaching experiments, ISI imprinting is 3D and immune to aberrations. Depletion of PP1-87B fails to cause significant changes in the stability of kinetochore-

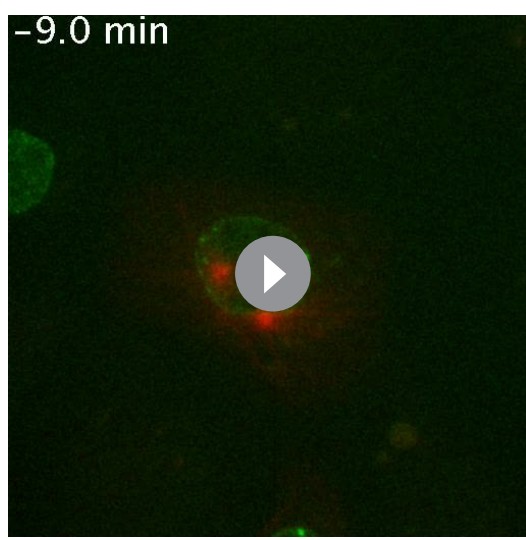

−9.0 min

**Video 1.** Mitotic progression of *Drosophila* S2 cells expressing Mad1-EGFP and mCherry-$\alpha$-Tubulin (related to *Figure 1A,B* and *Figure 1—figure supplement 1*). Mitotic progression of S2 cells co-expressing mCherry-$\alpha$-Tubulin (red) and Mad1-EGFP (green) was monitored by spinning disk confocal microscopy. Each frame represents a maximal intensity projection acquired every 60 s. NEB corresponds to time 0:0. Time is shown minutes.

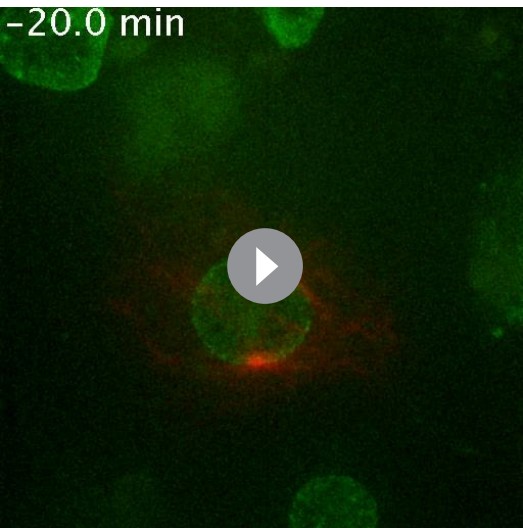

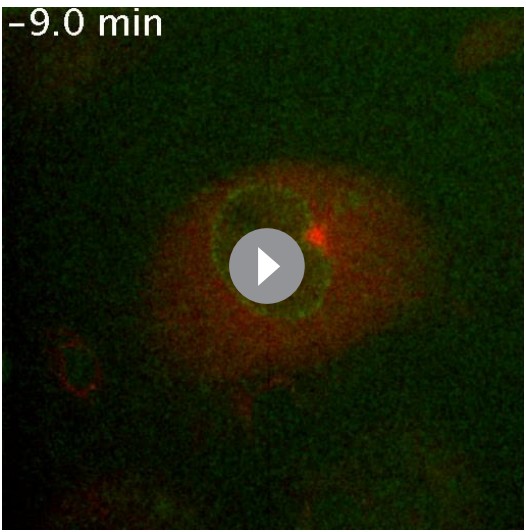

**Video 2.** Mitotic progression of PP1-87B depleted *Drosophila* S2 cells expressing Mad1-EGFP and mCherry-α-Tubulin (related to *Figure 1A,B* and *Figure 1—figure supplement 1*). Mitotic progression of S2 cells depleted of PP1-87B co-expressing mCherry-α-Tubulin (red) and Mad1-EGFP (green) was monitored by spinning disk confocal microscopy. Video illustrates the prolonged metaphase duration with Mad1-EGFP present at kinetochores of bioriented chromosomes in PP1-87B depleted cells. Each frame represents a maximal intensity projection acquired every 60 s. NEB corresponds to time 0:0. Time is shown minutes.

**Video 3.** Mitotic progression of PP1-87B and Mps1 depleted *Drosophila* S2 cells expressing Mad1-EGFP and mCherry-α-Tubulin (related to *Figure 1A,B* and *Figure 1—figure supplement 1*). Mitotic progression of S2 cells co-depleted of PP1-87B and Mps1 co-expressing mCherry-α-Tubulin (red) and Mad1-EGFP (green) was monitored by spinning disk confocal microscopy. Video illustrates that Mps1 depletion abolishes the prolonged metaphase delay and Mad1-EGFP accumulation at bioriented kinetochores observed in PP1-87B depleted cells. Each frame represents a maximal intensity projection acquired every 60 s. NEB corresponds to time 0:0. Time is shown minutes.

microtubule attachments in metaphase cells and there are no significant differences in the fractions of microtubules in the stable (KT-MT) versus unstable (non-KT-MT) populations when compared to control cells (*Figure 1H,I*; *Figure 1—figure supplement 2* and *Videos 4* and *5*). Taken together, these results clearly show that Mad1 accumulation at metaphase-aligned kinetochores and the delayed anaphase onset observed in PP1-87B-depleted cells are not caused by unattached kinetochores transiently generated as a result of increased Aurora B activity.

## PP1-87B antagonizes Mps1 T-loop autophosphorylation at kinetochores

The results described in the previous section suggest that the metaphase delay evoked by PP1-87B depletion results from problems in SAC silencing. Although, Mps1 is dispensable for kinetochore localization of Bub1 and BubR1 in flies (*Schittenhelm et al., 2009*; *Conde et al., 2013*) its activity is nonetheless required for Mad1 and Mad2 recruitment and therefore for SAC signaling (*Althoff et al., 2012*; *Conde et al., 2013*). In accordance, co-depletion of Mps1 with PP1-87B abolished Mad1 kinetochore recruitment and led to a premature anaphase onset, demonstrating that Mps1 is required for the SAC-dependent metaphase delay observed in PP1-87B RNAi (*Figure 1A, B*). Therefore, we sought to examine Mps1 kinetochore recruitment and activation in S2 cells depleted of PP1-87B. Immunofluorescence analysis showed that depletion of PP1-87B has no significant impact on Mps1 kinetochore localization. Mps1 accumulates at prometaphase kinetochores and its levels decrease significantly as microtubules attach, with a fraction of Mps1 persisting at kinetochores of metaphase chromosomes (*Figure 2A–D*). This residual pool of Mps1 is retained at bioriented kinetochores until anaphase onset, even after establishment of inter-kinetochore tension, as revealed by live imaging of S2 cells expressing EGFP-Mps1 under control of *Mps1 cis*-regulatory

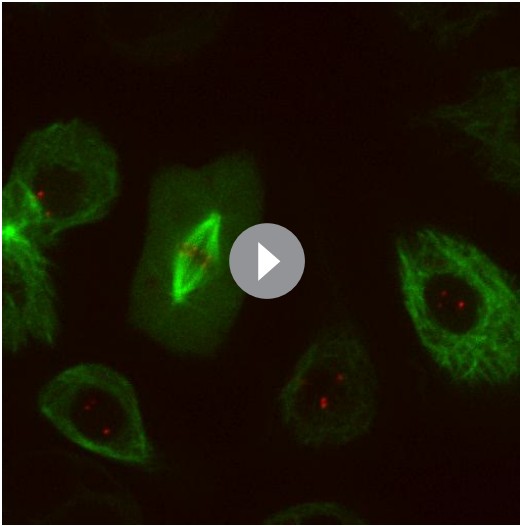

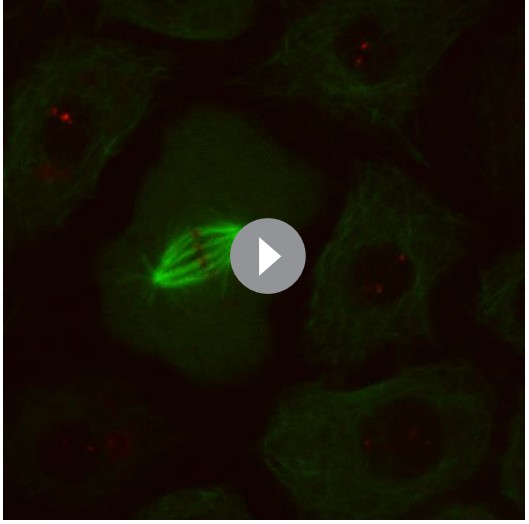

**Video 4.** Inducible speckle imaging (ISI) in a metaphase spindle of a control S2 cell expressing GFP-α-Tubulin and CID-mCherry (related to *Figure 1H,I* and *Figure 1—figure supplement 2*). The Video illustrates ISI imprinting and progressive contrast fade-out of GFP–α-tubulin in a control metaphase spindle. The observed speckle contrast fadeout of tubulin is due to microtubule turnover. The time between frames is variable and is presented on the corresponding excel source data file.

**Video 5.** Inducible speckle imaging (ISI) in a metaphase spindle of a PP1-87B depleted S2 cell expressing GFP-α-Tubulin and CID-mCherry (related to *Figure 1H,I* and *Figure 1—figure supplement 2*). The Video illustrates ISI imprinting and progressive contrast fade-out of GFP–α-tubulin in a PP1-87B depleted metaphase spindle. The observed speckle contrast fadeout of tubulin is due to microtubule turnover. The time between frames is variable and is presented on the corresponding excel source data file.

region (*Figure 2—figure supplement 1A,B* and *Video 6*). To evaluate Mps1 activation *status* we used a phospho-specific antibody that recognizes the conserved activating autophosphorylation (T490Ph) on Mps1 T-loop (*Jelluma et al., 2008*). Interestingly, we found that depletion of PP1-87B resulted in a substantial increase of Mps1 T-loop autophosphorylation during prometaphase and on metaphases kinetochores stably attached to microtubule bundles (*Figure 2A–D*, *Figure 2—figure supplement 1C,D* and *Figure 2—figure supplement 2A–F* with phospho-antibody characterization described in supplementary information). Moreover, depletion of PP1 noncatalytic subunit Sds22/PPP1R7, which positively regulates the phosphatase activity at kinetochores (*Posch et al., 2010*; *Wurzenberger et al., 2012*; *Heroes et al., 2013*), mimicked PP1-87B RNAi by increasing Mps1 T490 phosphorylation both at unattached and aligned kinetochores (*Figure 2—figure supplement 3A–D*). Mps1 association with prometaphase kinetochores in human cells is highly transient (*Jelluma et al., 2010*). Likewise, *Drosophila* Mps1 exhibits a fast turnover rate at unattached kineto-chores that relies in part on its kinase activity, as revealed by FRAP measurements of wild type (WT) and kinase-dead (KD) versions of EGFP-Mps1 expressed in S2 cells (*Figure 2E,F*). In line with the observed reduction in Mps1 T490 autophosphorylation, both the half-life and stable population of Mps1 increase at metaphase kinetochores (*Figure 2E,F*). However, in the absence of PP1-87B, Mps1 displays a faster dynamic exchange at aligned kinetochores, as expected from enhanced kinase activity (*Figure 2E,F*). Collectively, these results demonstrate that PP1-87B activity is required to represses the activating autophosphorylation of Mps1 that remains associated with metaphase kinetochores.

### PP1-87B/ PP1-γ dephosphorylates Mps1 T-loop

To evaluate whether PP1 directly antagonizes Mps1 T-loop activating autophosphorylation, we started by assessing *in vitro* the capacity of PP1-γ to dephosphorylate the kinase T-loop. Simultaneous incubation of PP1-γ with recombinant versions of human (Mps1/TTK) and *Drosophila* Mps1 orthologues led to a concentration-dependent decrease of T676 (T676Ph) and T490 (T490Ph)

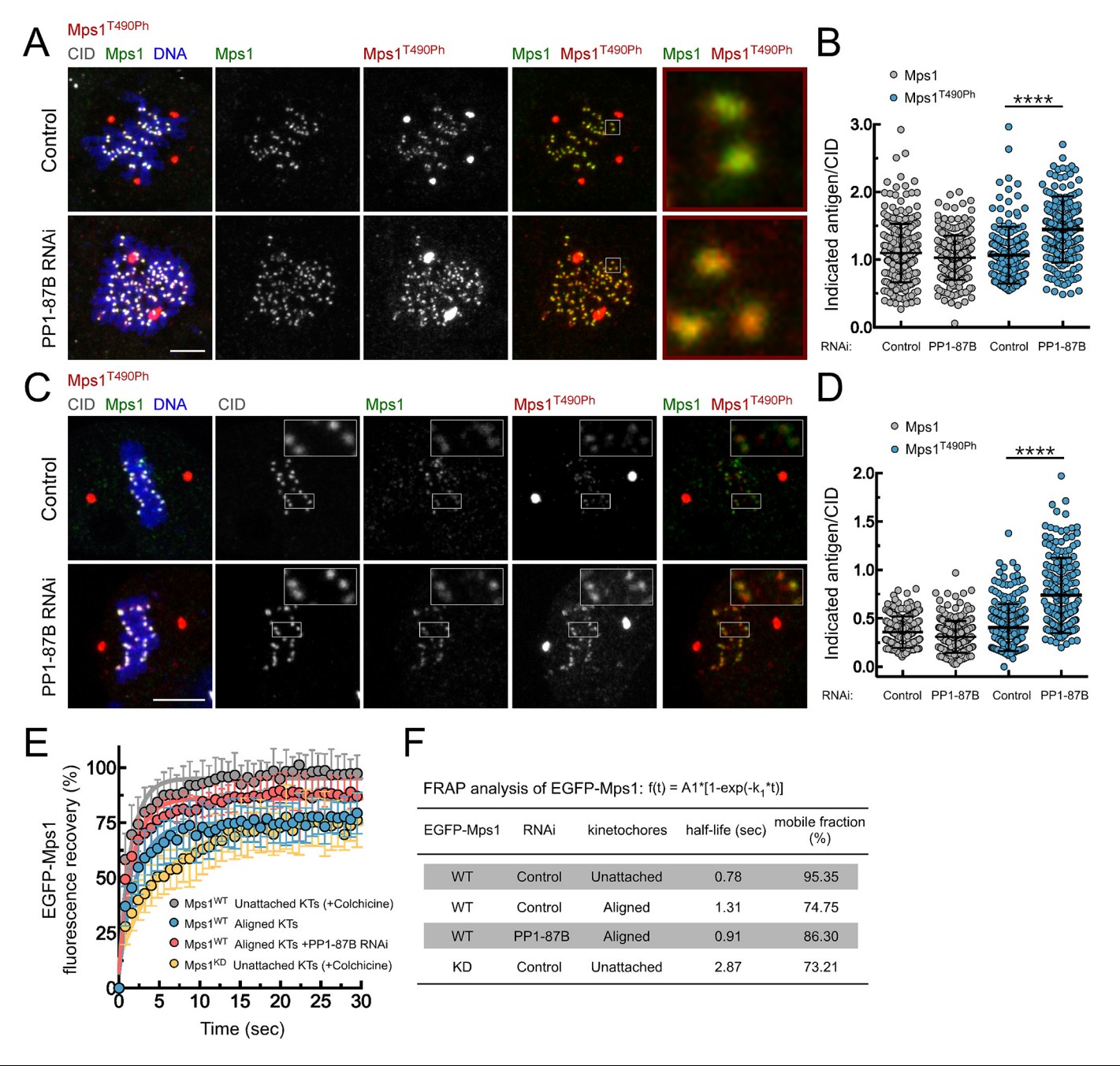

**Figure 2.** PP1-87B antagonizes Mps1 T-loop autophosphorylation at kinetochores of prometaphase and metaphase cells. (**A,B**) Representative immunofluorescence images (**A**) and corresponding quantifications (**B**) of Mps1 T490 phosphorylation (Mps1T490Ph) and Mps1 relative levels at prometaphase kinetochores of control and PP1-87B depleted *Drosophila* S2 cells. The insets display magnifications of the outlined regions. Mps1T490Ph and Mps1 fluorescence intensities were determined relative to CID signal (N ≥ 252 kinetochores from at least 10 cells for each condition). Scale bar: 5 µm. (**C,D**) Representative immunofluorescence images (**C**) and corresponding quantifications (**D**) of Mps1 T490 phosphorylation (Mps1T490Ph) and Mps1 relative levels at metaphase kinetochores of control and PP1-87B depleted S2 cells. The insets display magnifications of the outlined regions. Mps1T490Ph and Mps1 fluorescence intensities were determined relative to CID signal (N ≥ 217 kinetochores from at least 15 cells for each condition). Scale bar: 5 µm. (**E,F**) FRAP analysis of EGFP-Mps1WT (wild type) and EGFP-Mps1KD (kinase dead) at unattached or metaphase-aligned kinetochores of control and PP1-87B depleted S2 cells. To generate unattached kinetochores, S2 cells were treated with colchicine (30 µM) for 2 hr prior to FRAP experiments (**E**) Graph displays recovery-time curves of EGFP-Mps1 fluorescence (lines) and their time point means (dots) after bleaching of individual kinetochores in the indicated conditions. (**F**) Table showing the recovery half-lives and the % of mobile EGFP-Mps1 population obtained after fitting the average fluorescence intensity at each time point to a single-exponential one-phase association curve (N ≥ 9 cells for each condition). Data information: *Figure 2 continued on next page*

*Figure 2 continued*

in (B), (D) and (E) data are presented as mean ± SD. Asterisks indicate that differences between mean ranks are statistically significant, ****p<0.0001 (Mann-Whitney U test). Numerical source data for this figure are provided in *Figure 2—source data 1*.

The following source data and figure supplements are available for figure 2:

**Source data 1.** Numerical data for *Figure 2*.

**Figure supplement 1.** PP1-87B antagonizes the T-loop autophosphorylation of Mps1 that remains associated with metaphase kinetochores stably attached to spindle microtubules.

**Figure supplement 2.** Validation of a phospho-specific antibody against *Drosophila* Mps1 activating T-loop autophosphorylation.

**Figure supplement 3.** Depletion of Sds22 affects Mps1 T-loop autophosphorylation.

autophosphorylation, respectively (*Figure 3A,C*). Moreover, PP1-γ was able to dephosphorylate the T-loop of Mps1/TTK that had been previously activated by autophosphorylation (*Figure 3B*). In contrast, λ-phosphatase failed to antagonize to the same extent the autophosphorylation of human Mps1/TTK T676 and of Drosophila Mps1 T490 (*Figure 3C* and *Figure 3—figure supplement 1*), indicating a higher specificity of PP1-γ towards Mps1 T-loop *in vitro*.

We then sought to examine whether PP1 directly dephosphorylates Mps1 T-loop in a cellular context and determine its implications for SAC silencing. Because PP1 has numerous mitotic substrates acting on a multitude of pathways, its depletion results in pleiotropic phenotypes that preclude a clear and direct analysis of individual events. To circumvent this, we devised a strategy to specifically prevent Mps1 dephosphorylation without impacting the remaining PP1-87B cellular activity. The association of PP1 with the majority of its interactors is mediated by short motifs that bind to the phosphatase hydrophobic groove (*Egloff et al., 1997*; *Wakula et al., 2003*). *Drosophila* Mps1 N-terminus contains a clear match for a PP1-docking motif of the [R/K]VxF – type (*Figure 3D*). To assess the relevance of Mps1 KVLF$^{231-234}$ for the interaction with PP1-87B, we converted K231 and F234 to alanine and expressed both the wild type (EGFP-Mps1$^{WT}$) and mutant versions (EGFP-Mps1$^{K231A/F234A}$) of Mps1 in S2 cells. The interaction between Mps1 and PP1-87B was confirmed in pull-down assays in which EGFP-Mps1$^{WT}$ from S2 cell lysates bound efficiently to recombinant MBP-PP1-87B but not to the MBP used as negative control (*Figure 3E*). Conversely, mutating Mps1 KVLF$^{231-234}$ to AVLA$^{231-234}$ markedly decreased the ability of EGFP-Mps1 to interact with MBP-PP1-87B, hence validating KVLF as a PP1-binding motif (*Figure 3E,F*). To demonstrate that PP1-87B directly interacts with Mps1 via the KVLF motif, we performed pull-down assays with MBP-PP1-87B and bacterially purified fragments of Mps1 N-terminus (104–330 amino acids) harboring the wild-type (6xHis-N-Mps1$^{WT}$) or mutated PP1-docking motif (6xHis-N-Mps1$^{K231A/F234A}$). While 6xHis-N-Mps1$^{WT}$ bound efficiently to MBP-PP1-87B, the fragment 6xHis-N-Mps1$^{K231A/F234A}$ did not (*Figure 3G*), thus confirming that the KVLF motif mediates a direct interaction between Mps1 N-terminus and PP1-87B. In accordance, *in vitro* assays with EGFP-Mps1 immunoprecipitated from mitotic S2 cell lysates showed that recombinant PP1-γ antagonized with discernible less efficiency the T-loop autophosphorylation of EGFP-Mps1$^{K231A/F234A}$ in comparison to EGFP-Mps1$^{WT}$ (*Figure 3H,I*). Collectively, these results indicate that Mps1 T-loop is a direct substrate of PP1-87B/PP1-γ. However, we found that MBP-PP1-87B failed to pull-down EGFP-Mps1$^{WT}$ from S2 cells depleted of Sds22/PPP1R7 (*Figure 3E,F*), which suggests that in addition to the KVLF motif on the kinase N-terminus, the Sds22/PPP1R7 regulatory subunit is also required for competent binding of full-length Mps1 to PP1-87B in cell extracts.

To analyse the impact of diminished PP1-87B binding on cellular Mps1 activation and SAC signaling, we monitored Mps1 T-loop phosphorylation and mitotic progression in S2 cells expressing EGFP-Mps1$^{WT}$ and EGFP-Mps1$^{K231A/F234A}$ under control of *Mps1 cis*-regulatory region (*Figure 3—figure supplement 2*). Phosphorylation of T490 at unaligned kinetochores was significantly higher for EGFP-Mps1$^{K231A/F234A}$ relative to EGFP-Mps1$^{WT}$ (*Figure 3J,L*). A similar increment in the T-loop activation status was observed for EGFP-Mps1$^{WT}$ of prometaphase cells depleted of PP1-87B (*Figure 3J,L*). Notably, the mutation KVLF$^{231-234}$ prevented the dephosphorylation of Mps1 T-loop also at bioriented kinetochores, mimicking the result obtained in metaphase cells expressing EGFP-

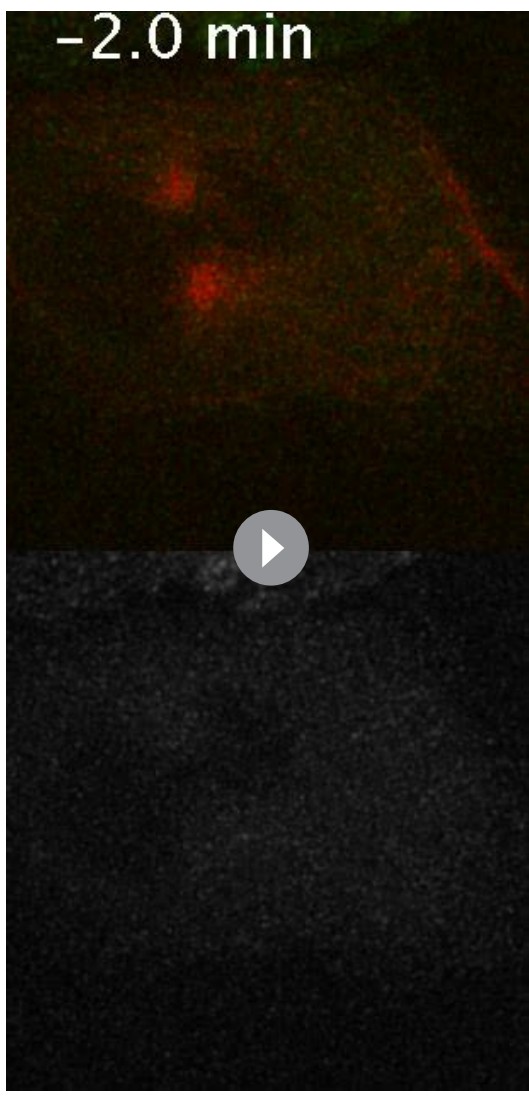

**Video 6.** Mitotic progression of *Drosophila* S2 cells expressing EGFP-Mps1$^{WT}$ and mCherry-α-Tubulin (related to *Figure 2—figure supplement 1A,B*). Mitotic progression of S2 cells expressing mCherry-α-Tubulin (red) and EGFP-Mps1$^{WT}$ (green) monitored by spinning disk confocal microscopy. Each frame represents a maximal intensity projection acquired every 2 min. NEB corresponds to time 0:0. Time is shown minutes.

Mps1$^{WT}$ in the absence of PP1-87B (*Figure 3K,L*). As expected from increased T490 autophosphorylation, EGFP-Mps1$^{K231A/F234A}$ exhibits faster dynamics than EGFP-Mps1$^{WT}$ at aligned kinetochores (*Figure 3—figure supplement 3A,B*). From these results, we conclude that PP1-87B directly controls Mps1 T-loop dephosphorylation at kinetochores during prometaphase and metaphase.

## PP1-mediated dephosphorylation of Mps1 T-loop inactivates the kinase and promotes SAC silencing

Live-cell imaging revealed that expression of EGFP-Mps1$^{K231A/F234A}$ caused a pronounced metaphase delay when compared to EGFP-Mps1$^{WT}$ expressing cells (*Figure 4A,B*; *Videos 7* and *8*). Depletion of BubR1 prevented this delay showing it is SAC-dependent (*Figure 4A,B* and *Video 9*). However, the delay is not caused by unstable kinetochore-microtubule interactions, since the inter-kinetochore distances monitored throughout metaphase and the attachments of bioriented kinetochores to cold-resistant microtubule bundles in EGFP-Mps1$^{K231A/F234A}$ cells were similar to those observed in metaphase cells expressing the wild-type kinase (*Figure 4C,D* and *Figure 4—figure supplement 1A*). Nevertheless, cells expressing EGFP-Mps1$^{K231A/F234A}$ retained higher levels of Mad1 at aligned kinetochores (*Figure 4E,F* and *Figure 4—figure supplement 1B,C*), which is consistent with lasting Mps1 activation and SAC signaling in metaphase. These results strongly suggest that impairing PP1-mediated dephosphorylation of the residual pool of Mps1 that remains associated with kinetochores of bioriented chromosomes is sufficient to delay SAC silencing and prevent anaphase onset despite normal chromosome attachment and high inter-kinetochore tension.

To further confirm the increment in Mps1 activity in EGFP-Mps1$^{K231A/F234A}$ and its impact on SAC signaling, we treated cells with colchicine to generate unattached kinetochores and monitored their capacity to arrest in mitosis. As expected for SAC competent cells, this led to an evident increase in the mitotic index of both EGFP-Mps1$^{WT}$ and EGFP-Mps1$^{K231A/F234A}$ expressing cells (*Figure 4G*). Consistent with the role of Aurora B in potentiating Mps1 activation and SAC signaling, inhibition of the former with increasing concentrations of Binucleine 2 (*Smurnyy et al., 2010*) weakened the SAC in EGFP-Mps1$^{WT}$ cells, as revealed by the gradual reduction in the mitotic index of colchicine incubated cultures (*Figure 4G*). In accordance, Aurora B inhibition negatively affected Mps1 T-loop autophosphorylation and Mad1 accumulation at unattached kinetochores. Strikingly, expression of EGFP-Mps1$^{K231A/F234A}$ partially rescued Mps1 T490 phosphorylation, Mad1 kinetochore recruitment and consistently attenuated the decline of SAC function following Aurora B inhibition (*Figure 4G–I*). Collectively, these results demonstrate that compromising Mps1 T-loop dephosphorylation is sufficient to delay SAC silencing and

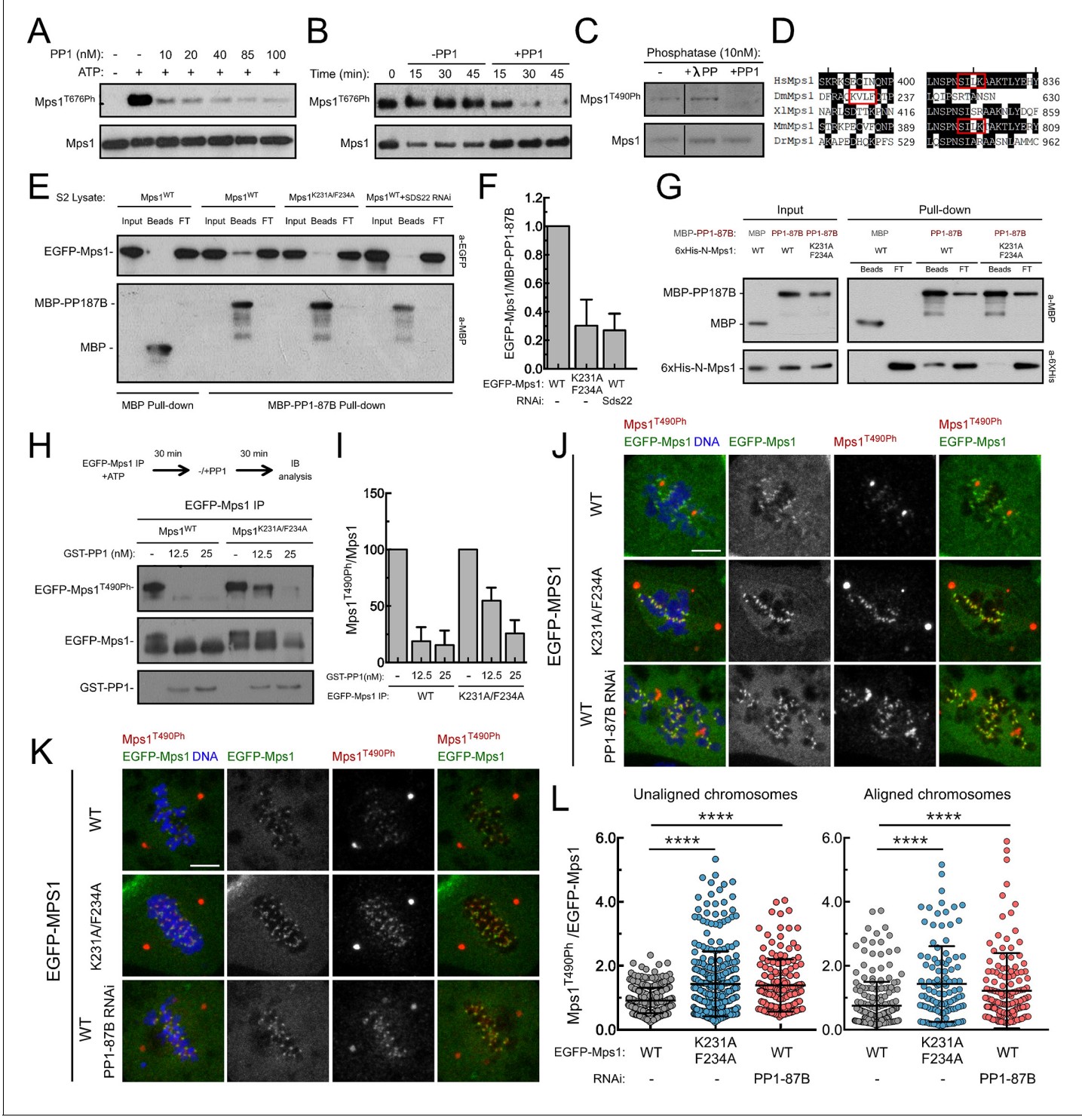

**Figure 3.** PP1-87B/PP1-γ dephosphorylates Mps1 T-loop *in vitro* and *in vivo*. (**A,B**) Western blot analysis of Mps1/TTK T676 autophosphorylation. Recombinant human Mps1 was incubated during 30 min with increasing concentrations of recombinant human PP1-γ (**A**) or PP1-γ (42.7 nM) was added to previously active Mps1 and phosphorylation of Mps1 T676 (Mps1^T676Ph^) monitored over time (**B**). (**C**) Western blot analysis of *Drosophila* Mps1 T490 autophosphorylation. Recombinant *Drosophila* Mps1 was incubated during 30 min with 10 nM of recombinant human PP1-γ or λ-PP and phosphorylation of Mps1 T490 (Mps1^T490Ph^) monitored. (**D**) Clustal W alignments of amino acid residues of indicated Mps1 orthologues. Red boxes highlight putative PP1-docking motifs identified in silico by the Eukaryotic Linear Motif (ELM) resource. (**E**) MBP-PP1-87B pull-downs from total cell lysates of S2 cells expressing EGFP-Mps1^WT^ or EGFP-Mps1^K231A/F234A^ or EGFP-Mps1^WT^ upon depletion of Sds22. Immobilized MBP was used as negative control. Input, beads and flow-through (FT) were probed by western blotting for the indicated proteins. (**F**) Quantification of Mps1 binding to

*Figure 3 continued on next page*

*Figure 3 continued*

PP1-87B from the pull-downs in (E). The chemiluminescence signal intensity of EGFP-Mps1 was determined relative to the signal of MBP-PP1-87B beads. The graph represents the quantification of relative levels of EGFP-Mps1 in MBP-PP1-87B pull-downs from at least two independent experiments. The values obtained for EGFP-Mps1$^{WT}$ from control cells were set to 1. (G) MBP-87B pull-downs of purified recombinant Mps1 N-terminus region (104–330 amino acids) harboring the wild-type (6xHis-N-Mps1$^{WT}$) or mutated PP1-docking motif (6xHis-N-Mps1$^{K231A/F234A}$). Immobilized MBP was used as negative control. Input, beads and flow-through (FT) were probed by western blotting for the indicated proteins. (H) Western blot analysis of EGFP-Mps1$^{WT}$ and EGFP-Mps1$^{K231A/F234A}$ T-loop dephosphorylation by PP1-γ. EGFP-Mps1$^{WT}$ and EGFP-Mps1$^{K231A/F234A}$ were immunoprecipitated from mitotic S2 lysates and incubated with ATP for 30 min to allow Mps1 T-loop autophosphorylation. Increasing concentrations of recombinant GST-PP1-γ were subsequently added to the reaction mixture and Mps1 T490 phosphorylation (Mps1$^{T490Ph}$) assessed after 30 min. Immunoprecipitates were probed by immunoblotting (IB) for the indicated proteins. (I) Quantification of Mps1$^{T490Ph}$ levels from the dephosphorylation assay in (H). The chemiluminescence signal intensity of Mps1$^{T490Ph}$ was determined relative to the corresponding signal of EGFP-Mps1 immunoprecipitates. The values obtained for each control reaction were set to 100%. The graph represents the quantification of Mps1$^{T490Ph}$ relative levels from two independent experiments. (J–L) Representative immunofluorescence images (J,K) and corresponding quantifications (L) of Mps1 T490 phosphorylation (Mps1T$^{T490Ph}$) at kinetochores of unaligned (J) and aligned (K) chromosomes from S2 cells expressing EGFP-Mps1$^{WT}$ or EGFP-Mps1$^{K231A/F234A}$ and from PP1-87B depleted S2 cells expressing EGFP-Mps1$^{WT}$. EGFP-Mps1 transgenes were expressed under the control of *Mps1 cis*-regulatory region (*Althoff et al., 2012*). Mps1T$^{T490Ph}$ fluorescence intensities were determined relative to EGFP-Mps1 signal (N ≥ 150 unaligned kinetochores from at least 8 cells for each condition and N ≥ 111 aligned kinetochores from at least 7 cells for each condition). Scale bar: 5 µm. Data information: in (F), (I) and (L) data are presented as mean ± SD. Asterisks indicate that differences between mean ranks are statistically significant, ****p<0.0001 (Kruskal-Wallis, Dunn's multiple comparison test). Numerical source data for this figure are provided in *Figure 3—source data 1*.

The following source data and figure supplements are available for figure 3:

**Source data 1.** Numerical data for *Figure 3*.
**Figure supplement 1.** Specificity of Mps1 T-loop dephosphorylation *in vitro*.
**Figure supplement 2.** Western blotting analysis of S2 cells expressing EGFP-Mps1$^{WT}$ and EGFP-Mps1$^{K231A/F234A}$.
**Figure supplement 3.** Mps1$^{WT}$ and Mps1$^{K231A/F234A}$ dynamics at kinetochores.

mitotic exit. Thus, we conclude that PP1-mediated dephosphorylation of Mps1 T-loop is required for efficient Mps1 inactivation and consequently for prompt SAC silencing.

## Preventing PP1-mediated inactivation of cytosolic Mps1 delays mitotic exit

Mps1 T-loop activation is first detected in the nucleoplasm and on pre-kinetochores during prophase (*Figure 5A*). Interestingly, kinetochore-null S2 cells engendered through CENP-C knockdown retain high levels of nucleoplasmic Mps1 T490 phosphorylation (*Figure 5A,B*), indicating that Mps1 activation at mitotic entry can occur through kinetochore-independent mechanisms. Since previous studies have shown that cytosolic Mps1 also contributes for SAC function (*Maciejowski et al., 2010*; *Maldonado and Kapoor, 2011*; *Rodriguez-Bravo et al., 2014*), we reasoned that the activation status of soluble Mps1 must be tightly controlled during mitosis to allow timely anaphase onset. This prompted us to examine whether PP1-87B also represses cytoplasmic Mps1 activation and whether this is relevant for SAC signaling. To assess the impact of PP1-mediated dephosphorylation specifically on cytosolic Mps1 and completely avert contributions from kinetochore-generated MCC, we monitored mitotic progression in CENP-C depleted cells (*Figure 5C*; *Figure 5—figure supplement 1* and *Videos 10–13*). Live-cell imaging revealed that in the absence of kinetochores, cells expressing EGFP-Mps1$^{WT}$ rapidly progressed through mitosis, whereas expression of EGFP-Mps1$^{K231A/F234A}$ markedly delayed anaphase onset (*Figure 5C,D*). Importantly, preventing MCC assembly through BubR1 co-depletion or expressing Mps1$^{K231A/F234A}$ harboring unphosphorylatable T490 (EGFP-Mps1$^{K231A/F234A/T490A}$), failed to cause an obvious increase in the mitotic timing indicating that the delay in mitosis observed in kinetochore-null Mps1$^{K231A/F234A}$ cells is SAC-dependent and requires Mps1 activating T-loop autophosphorylation (*Figure 5C,D*). These results show that impairing T-loop dephosphorylation of cytosolic Mps1 prolongs mitosis in a SAC-dependent manner even in the absence of a kinetochore-generated 'wait anaphase' signal. Therefore, timely transition to anaphase requires that SAC extinction at stably attached kinetochores coincides with the inactivation of

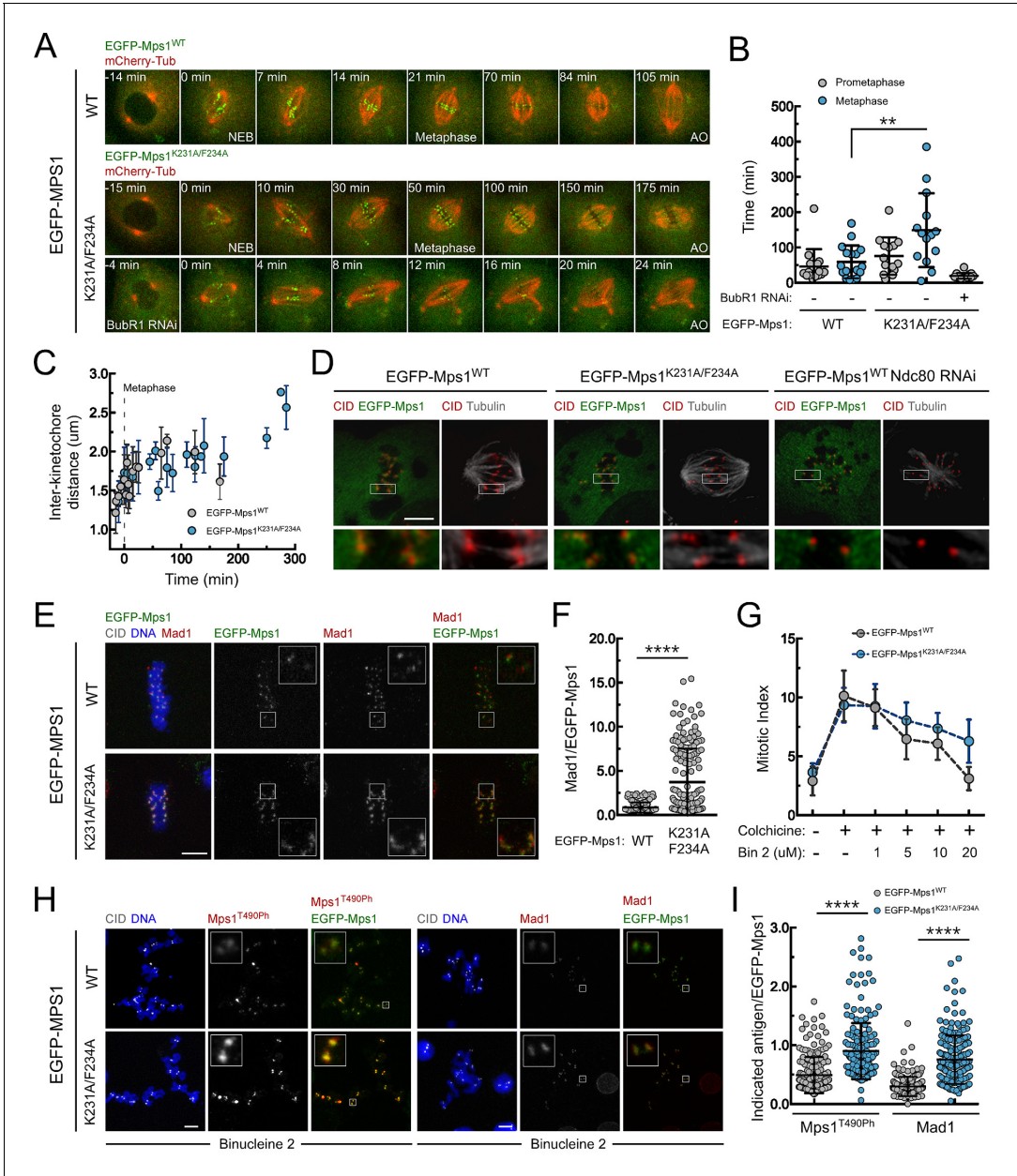

**Figure 4.** PP1-87B-mediated dephosphorylation of Mps1 T-loop renders the kinase inactive and is required for timely SAC silencing. (**A,B**) Mitotic progression (**A**) and prometaphase and metaphase duration (**B**) of S2 cells expressing mCherry-α-Tubulin and EGFP-Mps1$^{WT}$, EGFP-Mps1$^{K231A/F234A}$ or EGFP-Mps1$^{K231A/F234A}$ in a BubR1-depleted background. EGFP-Mps1 transgenes were expressed under control of *Mps1 cis*-regulatory region and mitotic progression monitored by time-lapse microscopy. The metaphase duration was defined by the length of time measured between the first still in which all kinetochore pairs were perfectly aligned at the metaphase plate (Metaphase) and anaphase onset (AO) (N ≥ 14 cells for each condition, from at least two independent experiments). (**C**) Quantification of inter-kinetochore distances throughout mitosis in S2 cells expressing EGFP-Mps1$^{WT}$ and EGFP-Mps1$^{K231A/F234A}$. Inter-kinetochore distances were measured as the distance between centroids of identified EGFP-Mps1 pairs at selected prometaphase and metaphase time frames. Time 0 min corresponds to the first metaphase frame. (**D**) Representative immunofluorescence images of cold-stable kinetochore fibers in S2 cells expressing EGFP-Mps1$^{WT}$, EGFP-Mps1$^{K231A/F234A}$ or EGFP-Mps1$^{WT}$ upon Ndc80 depletion. CID immunolocalization was used as kinetochore reference. The insets display magnifications of the outlined regions. Scale bar: 5 µm. (**E,F**) Representative immunofluorescence images (**E**) and corresponding quantifications (**F**) of Mad1 levels at metaphase kinetochores of S2 cells expressing EGFP-Mps1$^{WT}$ or EGFP-Mps1$^{K231A/F234A}$. Mad1 fluorescence intensities were determined relative to EGFP-Mps1 signal (N ≥ 162 kinetochores from at least 10 cells for each condition). Scale bar: 5 µm. (**G–I**) Mitotic index (**G**), representative immunofluorescence images (**H**) and quantification of Mps1T$^{490Ph}$ and Mad1 levels at unattached kinetochores (**I**) of colchicine treated S2 cells expressing EGFP-Mps1$^{WT}$ or EGFP-Mps1$^{K231A/F234A}$ in the presence of Aurora B inhibitor, binucleine 2 (Bin2) for 2 hr. Mitotic index in (**G**) was determined through p-H3 staining of cultured cells incubated in the absence or presence of colchicine (30 µM) for 12 hr. Increasing concentrations of Bin2 were added to cultures 10 hr after colchicine and mitotic index determined 2 hr later.
*Figure 4 continued on next page*

*Figure 4 continued*

In (**H** and **I**) Mps1T[490Ph] and Mad1 fluorescence intensities were determined relative to EGFP-Mps1 signal in cells treated with Bin2 (20 µM) and colchicine as in (**G**). Proteasome inhibitor MG132 (20 µM) was added to cultured cells 1 hr prior to Bin2 incubation to prevent mitotic exit (N ≥ 185 kinetochores from at least 9 cells for each condition). Scale bar: 5 µm. Data information: in (**B**), (**C**), (**F**), (**G**) and (**I**) data are presented as mean ± SD. Asterisks indicate that differences between mean ranks are statistically significant, \*\*p<0.05, \*\*\*\*p<0.0001 (Mann-Whitney U test). Numerical source data for this figure are provided in *Figure 4—source data 1*.

The following source data and figure supplement are available for figure 4:

**Source data 1.** Numerical data for *Figure 4*.

**Figure supplement 1.** Stable attachments and Mad1 levels at metaphase kinetochores of S2 cells expressing EGFP-Mps1[WT] and EGFP-Mps1[K231A/F234A].

cytosolic Mps1. These results allow us to conclude that PP1-mediated dephosphorylation of non-kinetochore Mps1 is a critical mechanism to efficiently silence the SAC and ensure prompt anaphase onset.

## Regulation of PP1 activity during mitosis

The activity of PP1 must be tightly regulated to allow robust Mps1 activation during early mitosis and to ensure its inactivation and SAC silencing when all kinetochores become stably attached to spindle microtubules. Immunofluorescence analysis of S2 cells revealed that PP1-87B preferentially accumulates on kinetochores when chromosomes are bioriented at the metaphase plate (*Figure 6A, B*). Kinetochore recruitment of PP1 in human cells is mediated in part by SILK and RVSF motifs present on KNL1 N-terminus. Aurora B-dependent phosphorylation of both motifs prevents the binding of PP1 during prometaphase (*Liu et al., 2010*). As *Drosophila* Spc105/KNL1 also contains a RVSF motif in its N-terminus, we examined whether a similar mechanism operates in S2 cells to control PP1-87B kinetochore localization. Cultured cells were incubated with Aurora B inhibitor Binucleine two in the presence of colchicine and of the proteasome inhibitor MG132, which respectively generate unattached kinetochores and prevent mitotic exit resulting from SAC abrogation. Inhibition of Aurora B led to an increase of PP1-87B levels at unattached kinetochores (*Figure 6C,D*), suggesting that similarly to the described for human cells, Aurora B activity limits the association of PP1-87B with unattached/unaligned kinetochores in flies. In addition to the control exerted by Aurora B on PP1 kinetochore localization, CDK1 phosphorylation of PP1 C-terminus was shown to repress the phosphatase activity during early mitosis (*Dohadwala et al., 1994*; *Yamano et al., 1994*; *Wu et al., 2009*; *Grallert et al., 2015*). In accordance, inhibition of CDK1 with RO-3306 in colchicine-treated S2 cells led to a reduction of Mps1 T490 autophosphorylation at unattached kinetochores, which was partially restored upon depletion of PP1-87B (*Figure 6E,F*). Thus, although CDK1 might also directly potentiate Mps1 activity (*Morin et al., 2012*), these results suggest that CDK1-dependent inhibition of PP1-87B contributes to maintain Mps1 active in prometaphase S2 cells. Declining Cyclin B levels were proposed

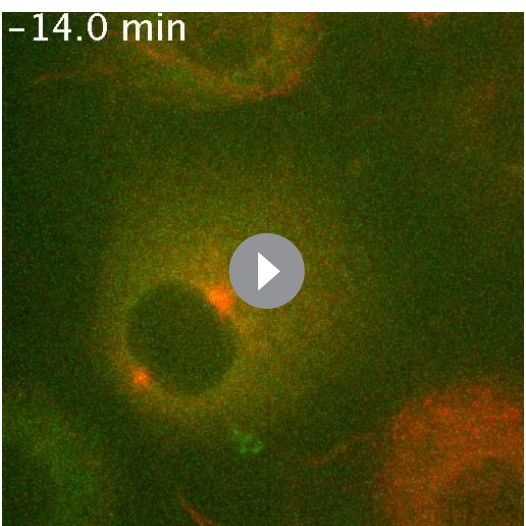

**Video 7.** Mitotic progression of *Drosophila* S2 cells expressing EGFP-Mps1[WT] and mCherry-α-Tubulin (related to *Figure 4A,B*). Mitotic progression of S2 cells expressing mCherry-α-Tubulin (red) and EGFP-Mps1[WT] (green) monitored by spinning disk confocal microscopy. Each frame represents a maximal intensity projection acquired every 7 min. NEB corresponds to time 0:0. Time is shown minutes.

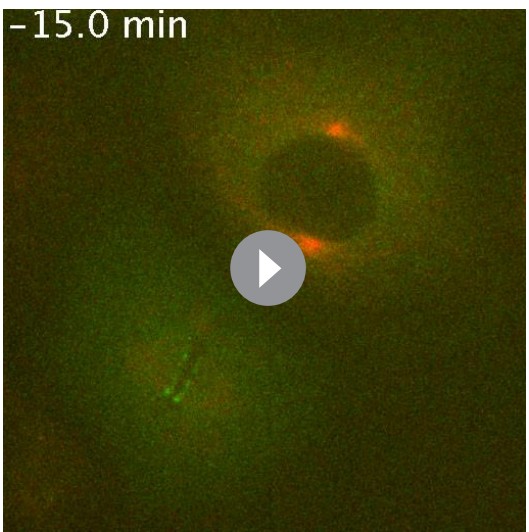

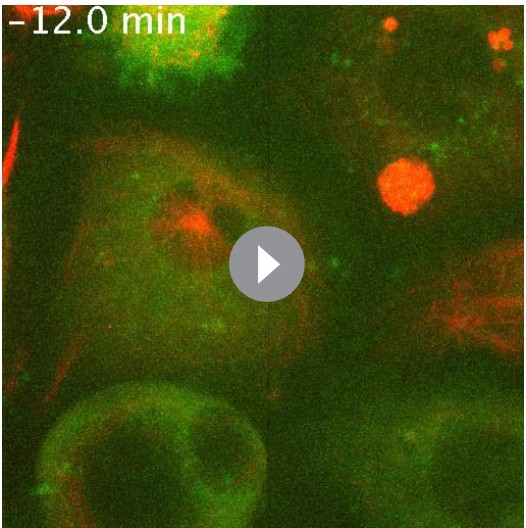

**Video 8.** Mitotic progression of *Drosophila* S2 cells expressing EGFP-Mps1$^{K231A/F234A}$ and mCherry-α-Tubulin (related to ***Figure 4A,B***). Mitotic progression of S2 cells expressing mCherry-α-Tubulin (red) and EGFP-Mps1$^{K231A/F234A}$ (green) monitored by spinning disk confocal microscopy. Video illustrates the prolonged metaphase delay. Each frame represents a maximal intensity projection acquired every 5 min. NEB corresponds to time 0:0. Time is shown minutes.

**Video 9.** Mitotic progression of BubR1-depleted *Drosophila* S2 cells expressing EGFP-Mps1$^{K231A/F234A}$ and mCherry-α-Tubulin (related to ***Figure 4A,B***). Mitotic progression of S2 cells depleted of BubR1 and expressing mCherry-α-Tubulin (red) and EGFP-Mps1 $^{K231A/F234A}$ (green) monitored by spinning disk confocal microscopy. Video illustrates that the prolonged metaphase duration caused by the expression of EGFP-Mps1$^{K231A/F234A}$ is SAC-dependent. Each frame represents a maximal intensity projection acquired every 4 min. NEB corresponds to time 0:0. Time is shown minutes.

to relieve PP1 inhibition, thereby ensuring high phosphatase activity in late mitosis (***Dohadwala et al., 1994***; ***Wu et al., 2009***; ***Grallert et al., 2015***). We monitored by live-cell imaging the levels of EGFP-Cyclin B throughout mitosis in S2 cells and as expected found that it starts to be rapidly degraded upon biorientation of all chromosomes (***Figure 6G,H*** and ***Video 14***). Interestingly, we could further observe that proteolysis of Cyclin B also occurs during prometaphase but at a significant lower rate (***Figure 6G,H*** and ***Video 14***). It was recently proposed that a small reduction in CDK1/Cyclin B activity is sufficient to allow PP1 auto-reactivation, which consequently triggers a feedback loop that ensures robust phosphatase activity (***Grallert et al., 2015***; ***Rogers et al., 2016***). Thus, the modest decline in Cyclin B that takes place before metaphase might be critical to permit PP1-mediated inactivation of Mps1 and SAC silencing.

## Discussion

How the SAC is rapidly extinguished allowing anaphase to proceed when the last chromosome becomes bioriented has remained a key unanswered question. The present study shows that efficient SAC silencing requires the inactivation of its most upstream regulator, Mps1 kinase. We demonstrate *in vitro* and *in vivo* that PP1 directly antagonizes Mps1 T-loop autophosphorylation rendering the kinase inactive at metaphase kinetochores and in the cytoplasm. In *Drosophila* this relies on a [R/K]VxF motif that is present on Mps1 N-terminus and requires the PP1 regulatory subunit Sds22. Importantly, Mps1 orthologues encompass several other PP1-binding motifs within different regions of the kinase, such as SILK signatures in the C-terminus of human and mouse Mps1. This suggests that despite the diversification in type and localization of PP1-docking motifs among Mps1 orthologues, the dephosphorylation of the kinase T-loop by PP1 is likely to be conserved.

The calponin homology (CH) domains of Ndc80 and Nuf2 directly mediate Mps1 accumulation at unattached kinetochores, which is proposed to promote the kinase activation required for efficient SAC signaling (***Kemmler et al., 2009***; ***Santaguida et al., 2011***; ***Saurin et al., 2011***; ***Nijenhuis et al.,***

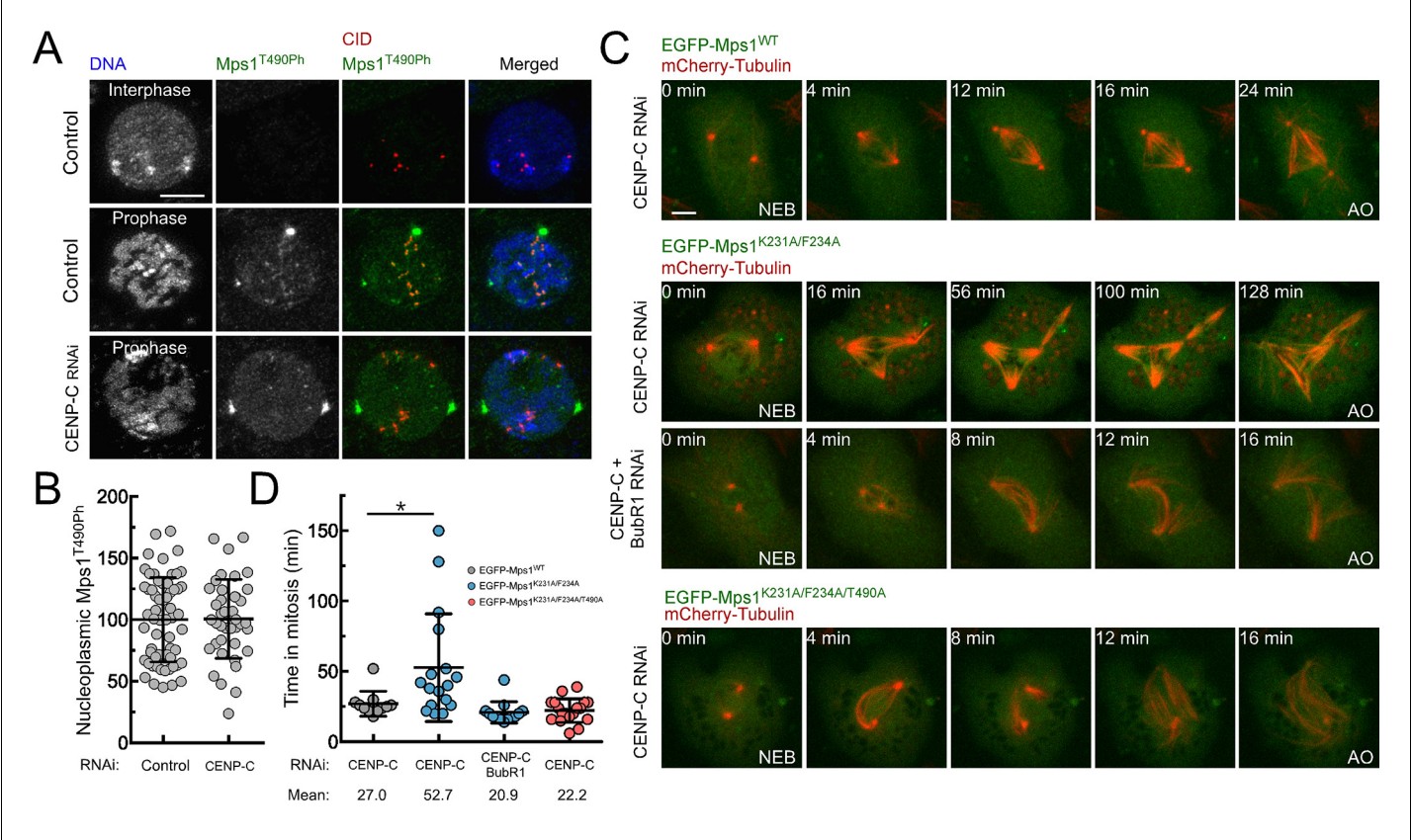

**Figure 5.** PP1-87B-mediated inactivation of cytosolic Mps1 is required for efficient SAC silencing and timely anaphase onset. (**A,B**) Representative immunofluorescence images (**A**) and corresponding quantification (**B**) of Mps1T490Ph levels in the nucleoplasm of control and CENP-C depleted cells in prophase. Mean values for control cells were set to 100% (N ≥ 40 cells for each condition from four independent experiments). Scale bar: 5 μm. (**C,D**) mitotic progression (**C**) and mitotic timing (**D**) of CENP-C depleted S2 cells expressing mCherry-Tubulin and EGFP-Mps1WT, EGFP-Mps1K231A/F234A or EGFP-Mps1K231A/F234A/T490A under control of *Mps1 cis*-regulatory region. Mitotic progression was monitored through time-lapse microscopy and the mitotic timing was defined by the length of time between nuclear envelope breakdown (NEB) and anaphase onset (AO) (N ≥ 11 cells for each condition from at least two independent experiments). Scale bar: 5 μm. Data information: in (**B**) and (**D**) data are presented as mean ± SD. Asterisks indicate that differences between mean ranks are statistically significant, *p<0.05 (Mann-Whitney U test). Numerical source data for this figure are provided in *Figure 5—source data 1*.

The following source data and figure supplement are available for figure 5:

**Source data 1.** Numerical data for *Figure 5*.

**Figure supplement 1.** Western blotting analysis of EGFP-Mps1WT, EGFP-Mps1K231A/F234A and EGFP-Mps1K231A/F234A/T490A expression in CENP-C depleted S2 cells.

*2013*). Because microtubules bind to the same surface of the CH domains that interacts with Mps1, microtubules forming end-on attachments outcompete Mps1 for Ndc80 binding (*Hiruma et al., 2015*; *Ji et al., 2015*). This competition mechanism is consistent with the reduction of Mps1 from metaphase kinetochores, but is however insufficient to dislodge all Mps1 molecules. As in human and budding yeast (*Vázquez-Novelle and Petronczki, 2010*; *Aravamudhan et al., 2015*), *Drosophila* kinetochores retain a fraction of Mps1 even after the formation of end-on attachments that exert proper kinetochore tension. Intriguingly, although low levels of Mps1 were shown to be sufficient to delay anaphase onset (*Herriott et al., 2012*; *Foijer et al., 2014*), the residual pool of Mps1 at metaphase kinetochores does not prevent SAC silencing. Work in budding yeast proposes that this is due to changes on kinetochore architecture imposed by end-on attachments. Stably bound microtubules exert mechanical tension on kinetochores, which increases the distance between the CH domains of

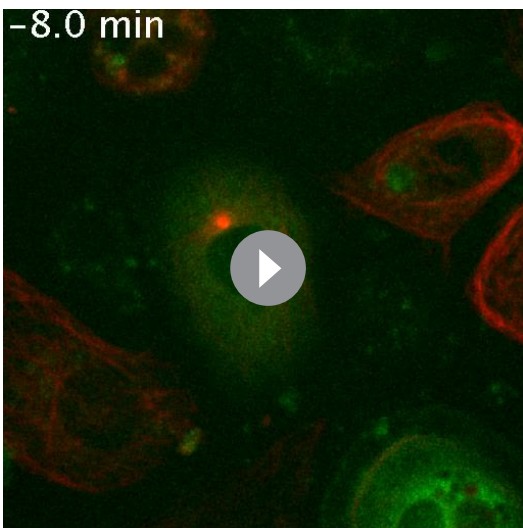

**Video 10.** Mitotic progression of CENP-C depleted *Drosophila* S2 cells expressing EGFP-Mps1[WT] and mCherry-α-Tubulin (related to *Figure 5C,D*). S2 cells depleted of CENP-C expressing mCherry-α-Tubulin (red) and EGFP-Mps1[WT] (green) monitored by spinning disk confocal microscopy. Each frame represents a maximal intensity projection acquired every 4 min. NEB corresponds to time 0:0. Time is shown minutes.

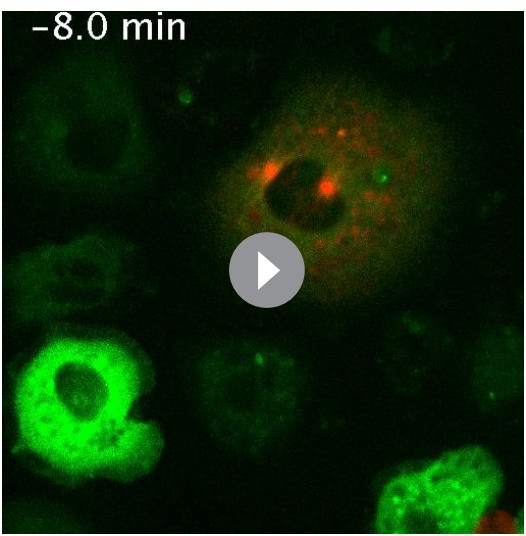

**Video 11.** Mitotic progression of CENP-C depleted *Drosophila* S2 cells expressing EGFP-Mps1[K231A/F234A] and mCherry-α-Tubulin (related to *Figure 5C,D*). S2 cells depleted of CENP-C expressing mCherry-α-Tubulin (red) and EGFP-Mps1[K231A/F234A] (green) monitored by spinning disk confocal microscopy. Video illustrates that impairing the interaction between Mps1 and PP1-87B delays mitotic exit in kinetochore-null cells. Each frame represents a maximal intensity projection acquired every 4 min. NEB corresponds to time 0:0. Time is shown minutes.

the Ndc80 complex and the MELT motifs on Spc105/KNL1, thus preventing phosphorylation of the latter by Mps1 and allowing SAC silencing (*Aravamudhan et al., 2015*). However, since in higher eukaryotes kinetochore-localized Mps1 dynamically exchanges with the cytoplasmic pool (*Jelluma et al., 2010*), free diffusion of Mps1 is expected to overcome the physical separation between the kinase receptor and its substrates at metaphase kinetochores. Moreover, the recruitment of Mad1-Mad2 to budding yeast kinetochores occurs exclusively through a linear pathway in which Mps1-phosphorylated MELTs function as docking sites for the binding of Bub1-Bub3 complexes, which in turn act as a loading platform for Mad1-Mad2 localization (*London and Biggins, 2014*). In *Drosophila* and in human cells however, an Spc105/KNL1- and Bub1-independent pathway also recruits Mad1-Mad2 to unattached kinetochores (*Schittenhelm et al., 2009*; *Caldas et al., 2015*; *Silió et al., 2015*). This relies on the localization of the RZZ complex at kinetochores, which is itself regulated by Mps1 activity (*Santaguida et al., 2010*) and is sufficient to activate the SAC in response to unattached kinetochores, even in the absence of KNL1 (*Schittenhelm et al., 2009*; *Silió et al., 2015*). Hence, in addition to the physical separation between the Ndc80 complex and Spc105/KNL1 phosphodomain, other mechanisms must operate in metazoans to limit the activity of kinetochore-associated Mps1 and extinguish SAC signaling in metaphase. Here we demonstrate that this is accomplished by PP1-mediated dephosphorylation of Mps1 T-loop. We found that depletion of PP1-87B or preventing its interaction with Mps1 increases the activation status of the residual fraction of Mps1 that persists at kinetochores of bioriented chromosomes. This correlates with higher levels of Mad1 retained on kinetochores and a prolonged metaphase delay that is caused by chronic SAC engagement despite stable end-on attachments and establishment of kinetochore tension. Thus, we reason that intra-kinetochore tension or alterations on kinetochore architecture, which also correlate with SAC satisfaction in *Drosophila* and human cells (*Maresca and Salmon, 2009*; *Uchida et al., 2009*), are important not only to disrupt the interaction of Mps1 with its kinetochore substrates but also to increase the recruitment of PP1 to metaphase chromosomes. Similarly to human cells (*Liu et al., 2010*), kinetochore levels of PP1-87B in *Drosophila* S2 cells are significantly

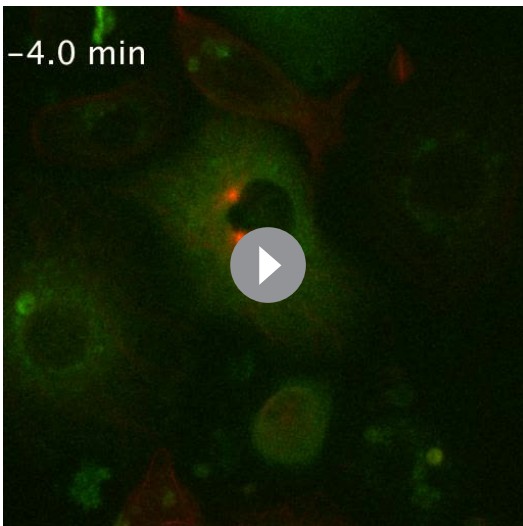
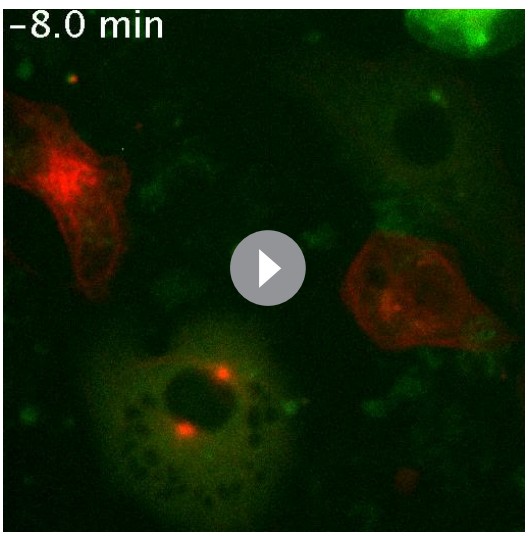

**Video 12.** Mitotic progression of *Drosophila* S2 cells co-depleted of CENP-C and BubR1 expressing EGFP-Mps1$^{K231A/F234A}$ and mCherry-α-Tubulin (related to ***Figure 5C,D***). S2 cells co-depleted of CENP-C and BubR1 expressing mCherry-α-Tubulin (red) and EGFP-Mps1$^{K231A/F234A}$ (green) monitored by spinning disk confocal microscopy. Video illustrates that the depletion of BubR1 from kinetochore-null cells expressing EGFP-Mps1$^{K231A/F234A}$ accelerates anaphase onset. Each frame represents a maximal intensity projection acquired every 4 min. NEB corresponds to time 0:0. Time is shown minutes.

**Video 13.** Mitotic progression of CENP-C depleted *Drosophila* S2 cells expressing EGFP-Mps1$^{K231A/F234A/T490A}$ and mCherry-α-Tubulin (related to ***Figure 5C,D***). S2 cells depleted of CENP-C expressing mCherry-α-Tubulin (red) and EGFP-Mps1$^{K231A/F234A/T490A}$ (green) monitored by spinning disk confocal microscopy. Video illustrates that the extended mitotic timing in kinetochore-null cells expressing EGFP-Mps1$^{K231A/F234A}$ requires Mps1 T-loop phosphorylation. Each frame represents a maximal intensity projection acquired every 4 min. NEB corresponds to time 0:0. Time is shown minutes.

higher at bioriented chromosomes. The association of PP1-γ with human kinetochores is directly mediated by SILK and [R/K]VxF motifs on Spc105/KNL1 N-terminus and by the Spindle- and Kinetochore-Associated (Ska) complex (*Sivakumar et al., 2016*). At tensionless kinetochores, the recruitment of PP1-γ is limited due to phosphorylation of Spc105/KNL1 N-terminus and of the Ska complex by Aurora B, which respectively inhibits the binding of PP1-γ to Spc105/KNL1 and Ska kinetochore localization (*Liu et al., 2010*; *Chan et al., 2012*). As microtubules bind to kinetochores and tension is established, the spatial repositioning of the KMN network decreases Aurora-B mediated phosphorylation of the Ska complex and of Spc105/KNL1 N-terminus (*Welburn et al., 2010*), hence allowing bulk accumulation of PP1-γ, which then catalyses the dephosphorylation of Mps1 T-loop. This provides a mechanism to ensure the complete inactivation of any vestigial Mps1 residing at microtubule-attached kinetochores during late stages of mitosis.

In addition to the MCC generated at unattached kinetochores in prometaphase, APC/C inhibitory complexes are also assembled at the nuclear pore and in the cytoplasm during late interphase and prophase (*Sudakin et al., 2001*; *Lopes et al., 2005*; *Lince-Faria et al., 2009*; *Maciejowski et al., 2010*; *Schweizer et al., 2013*; *Rodriguez-Bravo et al., 2014*). This pre-mitotic MCC defines the minimum length of time a cell will spend in mitosis as it ensures APC/C inhibition until newly assembled kinetochores are able to generate an efficient SAC response. Interestingly, inhibition of Mps1 causes a substantial decrease in pre-mitotic MCC levels and a consequent reduction in Cyclin B levels already in G2 and prophase (*Maciejowski et al., 2010*; *Rodriguez-Bravo et al., 2014*). Moreover, a truncated form of Mps1 unable to localize at kinetochores failed to promote the recruitment of Bub1 to unattached kinetochores but was nevertheless sufficient to sustain the assembly of mitotic APC/C inhibitory complexes and delay anaphase onset (*Maciejowski et al., 2010*). In addition, cytosolic Mps1 activity was shown to be critical to maintain the metaphase arrest caused by Mad1 constitutively tethered to stably attached bioriented kinetochores that were stripped from Mps1 (*Maldonado and Kapoor, 2011*). These observations indicate that the activity of soluble Mps1

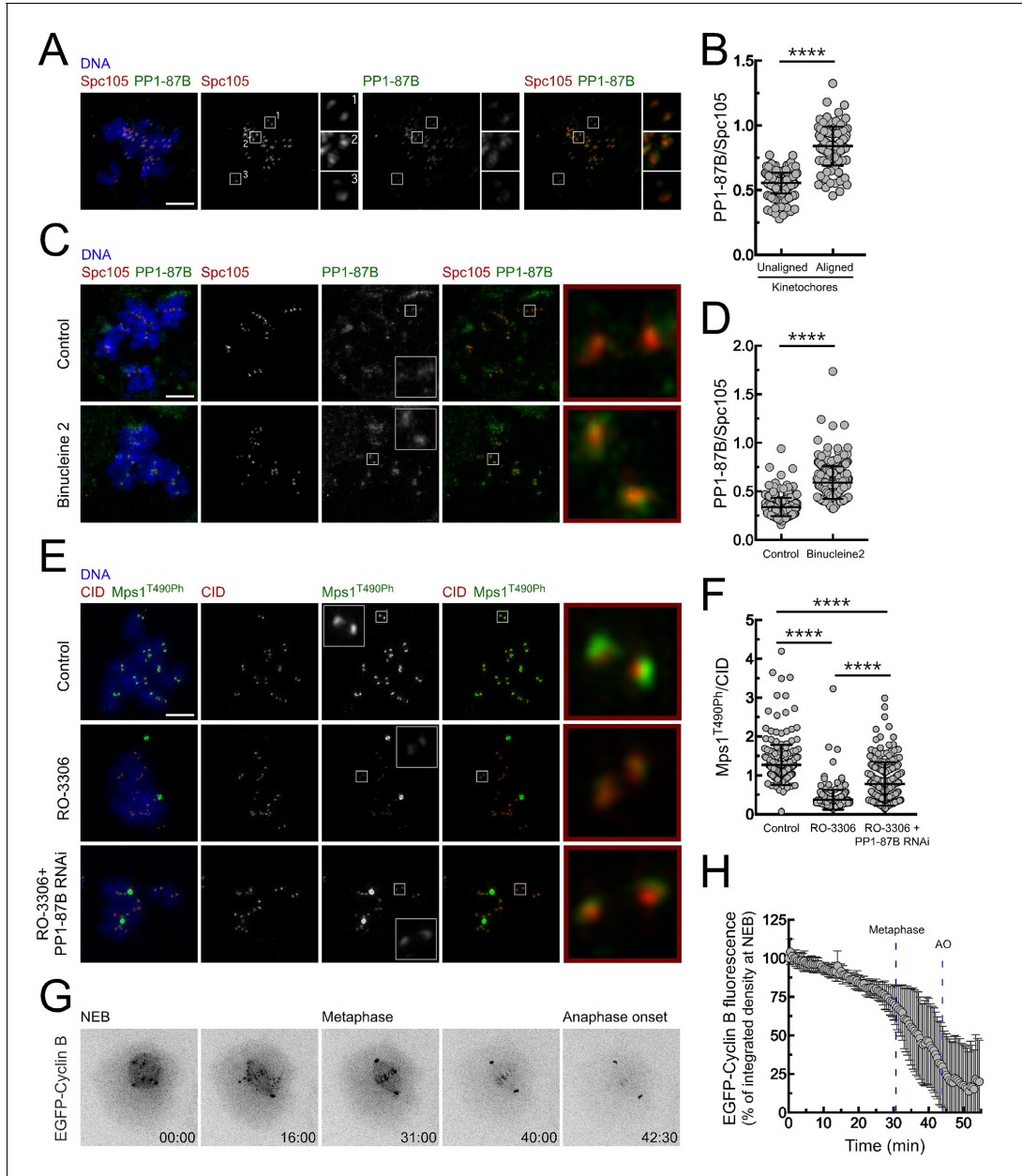

**Figure 6.** Regulation of PP1-87B activity during mitosis. (A,B) Representative immunofluorescence images (A) and corresponding quantifications (B) of PP1-87B levels at kinetochores of unaligned and aligned chromosomes in S2 cells. PP1-87B fluorescence intensities were determined relative to Spc105 signal (N = 241 unaligned kinetochores and N = 110 aligned kinetochores from at least 6 cells). Scale bar: 5 μm. (C,D) Representative immunofluorescence images (C) and corresponding quantifications (D) of PP1-87B levels at unattached kinetochores of control and Binucleine 2-treated S2 cells. Cells were treated with MG132 (20 μM) for 1 hr followed by a 2 hr incubation with colchicine (30 μM) to generate unattached kinetochores. Binucleine 2 (20 μM) was added to cultures 30 min prior to colchicine treatment. PP1-87B fluorescence intensities were determined relative to Spc105 signal (N ≥ 282 kinetochores from at least 12 cells for each condition). Scale bar: 5 μm. (E,F) Representative immunofluorescence images (E) and corresponding quantifications (F) of Mps1$^{T490Ph}$ levels at unattached kinetochores of control and PP1-87B depleted cells in the presence of CDK1 inhibitor (RO-3306, 10 μM) for 1 hr. To generate unattached kinetochores cultured cells were incubated with colchicine (30 μM) for 2 hr. The insets display magnifications of the outlined regions. Mps1T$^{490Ph}$ fluorescence intensities were determined relative to CID signal (N ≥ 236 kinetochores from at least 10 cells for each condition). Scale bar: 5 μm. (G,H) Representative mitotic progression (G) and Cyclin B degradation profile (H) of S2 cells during an unperturbed mitosis. Mitotic progression of S2 cells expressing EGFP-Cyclin B was monitored by time-lapse microscopy and the integrated intensity of EGFP-Cyclin B fluorescence measured in all the frames. The values for EGFP-Cyclin B fluorescence at nuclear envelope breakdown (NEB) (time 0 min) were set to 100%. Dashed blue lines indicate the mean time from NEB to metaphase and to anaphase onset (AO) (N = 5 cells from a single experiment). Data information: in (B), (D), (F) and (H) data are presented as mean ± SD. Asterisks indicate that differences between mean ranks are statistically significant, ****p<0.0001 (Mann-Whitney U test). Numerical source data for this figure are provided in *Figure 6—source data 1*.
*Figure 6 continued on next page*

*Figure 6 continued*

The following source data is available for figure 6:

**Source data 1.** Numerical data for *Figure 6*.

is required to generate the MCC and/or prevent its disassembly during mitosis, further strengthening the idea that removing the kinase from kinetochores is not sufficient for efficient SAC silencing. Therefore, determining how cytosolic Mps1 is inactivated after the SAC signal is extinguished at stably attached kinetochores becomes essential to understand the mechanism that allows timely mitotic exit. Here we show that PP1 directly controls Mps1 activation, not only at kinetochores but also in the cytoplasm. We demonstrate that preventing PP1-mediated dephosphorylation of soluble Mps1 maintains the kinase active in the cytosol and that this is sufficient to significantly delay mitotic exit even in the absence of kinetochore-generated MCC. Although kinetochore recruitment of Mps1 substantially increases the cell capacity to sustain a prolonged mitotic arrest when challenged with spindle poisons (*Nijenhuis et al., 2013*), during an unperturbed mitosis in which MCC production at kinetochores is minimal (*Collin et al., 2013*; *Dick and Gerlich, 2013*), controlled activity of soluble Mps1 is expected to be particularly relevant. We envisage that the inactivation of cytosolic Mps1 is critical to ensure SAC responsiveness at kinetochores during late mitosis so that transition to anaphase occurs without delay after the last unattached kinetochore forms stable attachments.

CDK1 phosphorylation of PP1 C-terminus represses the phosphatase activity during early mitosis (*Dohadwala et al., 1994*; *Yamano et al., 1994*; *Wu et al., 2009*; *Grallert et al., 2015*). Our data suggest that CDK1-dependent inhibition of PP1-87B contributes to maintain Mps1 active in prometaphase S2 cells. Declining Cyclin B levels were shown to relieve PP1 inhibition (*Dohadwala et al., 1994*; *Wu et al., 2009*; *Grallert et al., 2015*). However, several lines of evidence indicate that the SAC is switched off under condition of high CDK1/Cyclin B activity (*Mirchenko and Uhlmann, 2010*; *Oliveira et al., 2010*; *Kamenz and Hauf, 2014*; *Rattani et al., 2014*; *Vázquez-Novelle et al., 2014*), hence challenging the impact that CDK1/Cyclin B-mediated regulation of PP1 might have on SAC silencing. Interestingly, recent mathematical modelling support that small decreases in CDK1/Cyclin B activity are sufficient to initiate PP1 re-activation of and trigger a positive feed-back loop that ensures robust phosphatase activity (*Rogers et al., 2016*). Therefore, slow Cyclin

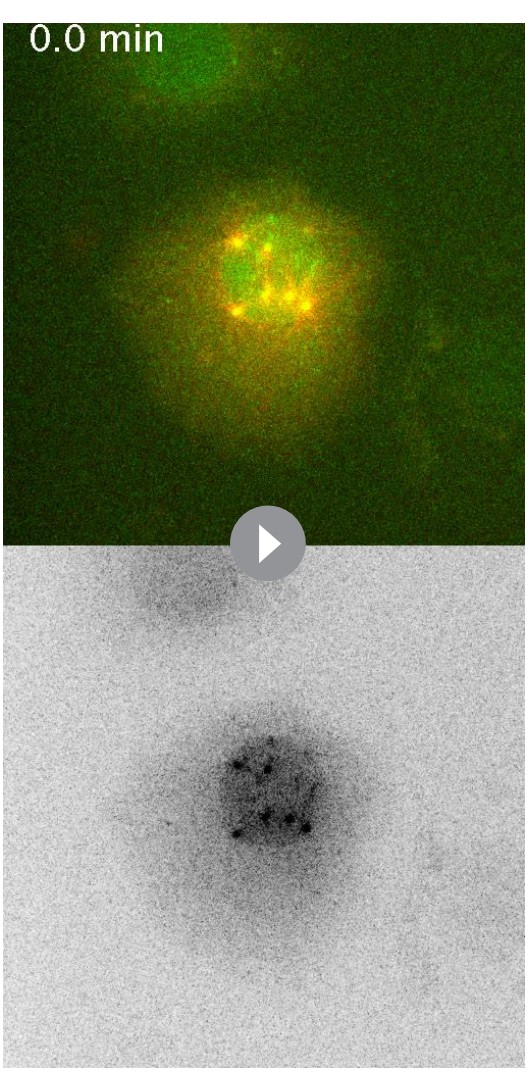

**Video 14.** Mitotic progression of *Drosophila* S2 cells expressing EGFP-Cyclin B. (related to *Figure 6G,H*). S2 cells expressing mCherry-α-Tubulin (red) and EGFP-Cyclin B (green) monitored by spinning disk confocal microscopy. Each frame represents a maximal intensity projection acquired every 30 s. NEB corresponds to time 0:0. Time is shown minutes.

B proteolysis occurring before metaphase in S2 cells might be sufficient to allow PP1-mediated inactivation of Mps1 and SAC silencing when biorientation of all chromosomes is achieved. Another possibility is that global repression of PP1 during early mitosis is ensured by CDK1/Cyclin A. Cyclin A is progressively degraded during prometaphase and its abundance governs the overall stability of kinetochore-microtubule attachments, which increases in metaphase when cyclin A levels fall below a critical threshold (*Kabeche and Compton, 2013*). It would be interesting to investigate whether decreased CDK1/Cyclin A coordinates this switch in attachment stability at the prometaphase to metaphase transition with the re-activation of PP1 and consequently with SAC silencing. This could explain how PP1 switches off the SAC despite elevated Cyclin B levels in early metaphase.

Taking our results together with previously reported work (*Dohadwala et al., 1994*; *Yamano et al., 1994*; *Liu et al., 2010*; *Mochida and Hunt, 2012*; *Grallert et al., 2015*; *Qian et al., 2015*; *Rogers et al., 2016*), we propose a model that controls Mps1 activation in a timely manner and perfectly coordinated with mitotic progression. Elevated CDK1 and Aurora B activities during prometaphase, respectively repress PP1 activity and avert the phosphatase localization at unattached kinetochores. This ensures high levels of active kinetochore- and cytosolic Mps1 and therefore, robust SAC signaling in early mitosis (*Figure 7*). As kinetochores become stably attached to spindle microtubules, Mps1 levels drastically decrease whereas PP1 accumulation increases. This occurs concomitantly with a drop in Cyclin B (and Cyclin A) that allows prompt PP1-mediated inactivation of Mps1 at kinetochores and in the cytoplasm (*Figure 7*). We propose that this coincides with the dephosphorylation of Mps1 substrates, with the removal of other SAC proteins and with modifications on kinetochore organization. As Mps1 promotes MCC assembly through multi-target phosphorylations along the SAC signaling cascade, the checkpoint is highly responsive to changes in Mps1 activity (*Faesen et al., 2017*; *Ji et al., 2017*). Inactivation of Mps1 by PP1 prevents the kinase from counteracting the dephosphorylation of its substrates, thus avoiding futile cycles and ensuring efficient and rapid SAC silencing. This guarantees a swift anaphase onset following chromosome biorientation.

## Materials and methods

### S2 cell cultures, RNAi and drug treatments

*Drosophila* S2 cell cultures (S2-DGRC), RNAi synthesis and RNAi treatments were performed as previously described (*Conde et al., 2013*). At selected time points, cells were collected and processed for immunofluorescence, time-lapse microscopy or immunoblotting. When required, cells were subjected to several drug treatments before being collected and processed for the desired analysis. In order to promote microtubule depolymerisation, cells were incubated with 30 µM colchicine (Sigma–Aldrich, St. Louis, MO) for 2–12 hr. To inhibit Aurora B activity, selected concentrations of Binucleine 2 (Sigma-Aldrich) were added to cultured cells for 2 hr. To inhibit CDK1/Cyclin B activity, 10 µM RO-3306 (Sigma-Aldrich) were added to cultured cells for 1 hr. When required the 20 µM MG132 (Calbiochem, San Diego, CA) was added to cultured cells to inhibit the proteasome. The *Drosophila* S2-DGRC cell line (stock#6) was acquired from the Drosophila Genomics Resource Center, Indiana University and was not independently authenticated. The cell lines were routinely tested negative for mycoplasma contamination.

### Constructs and S2 cell transfection

The pCaSpeR4 harboring EGFP-Mps1$^{WT}$ under control of *Mps1 cis*-regulatory region was a gift from Christian Lehner (*Althoff et al., 2012*). Constructs EGFP-Mps1$^{K231A/F234A}$ and EGFP-Mps1$^{K231A/F234A/T490A}$ were generated by site-directed mutagenesis with primers harboring the desired mutation. PCR reactions were performed with Phusion polymerase (New England Biolabs, Ipswich, MA) and pCaSpeR4-EGFP-Mps1$^{WT}$ as template. PCR products were digested with DpnI restriction enzyme (New England Biolabs), used to transform competent bacteria and selected for positives. To overexpress EGFP-Mps1$^{WT}$ and EGFP-Mps1$^{K231A/F234A}$ in S2 cells, Mps1 coding sequence was cloned in frame with N-terminal EGFP under regulation of a metallothionein promoter in the pMT-EGFP-C vector (Invitrogen, Carlsbad, CA) as previously described (*Conde et al., 2013*). pMT-EGFP-Mps1$^{K231A/F234A}$ was generated by site-directed mutagenesis with primers harboring the desired mutation. PCR reactions were performed with Phusion polymerase (New England Biolabs) and pMT-

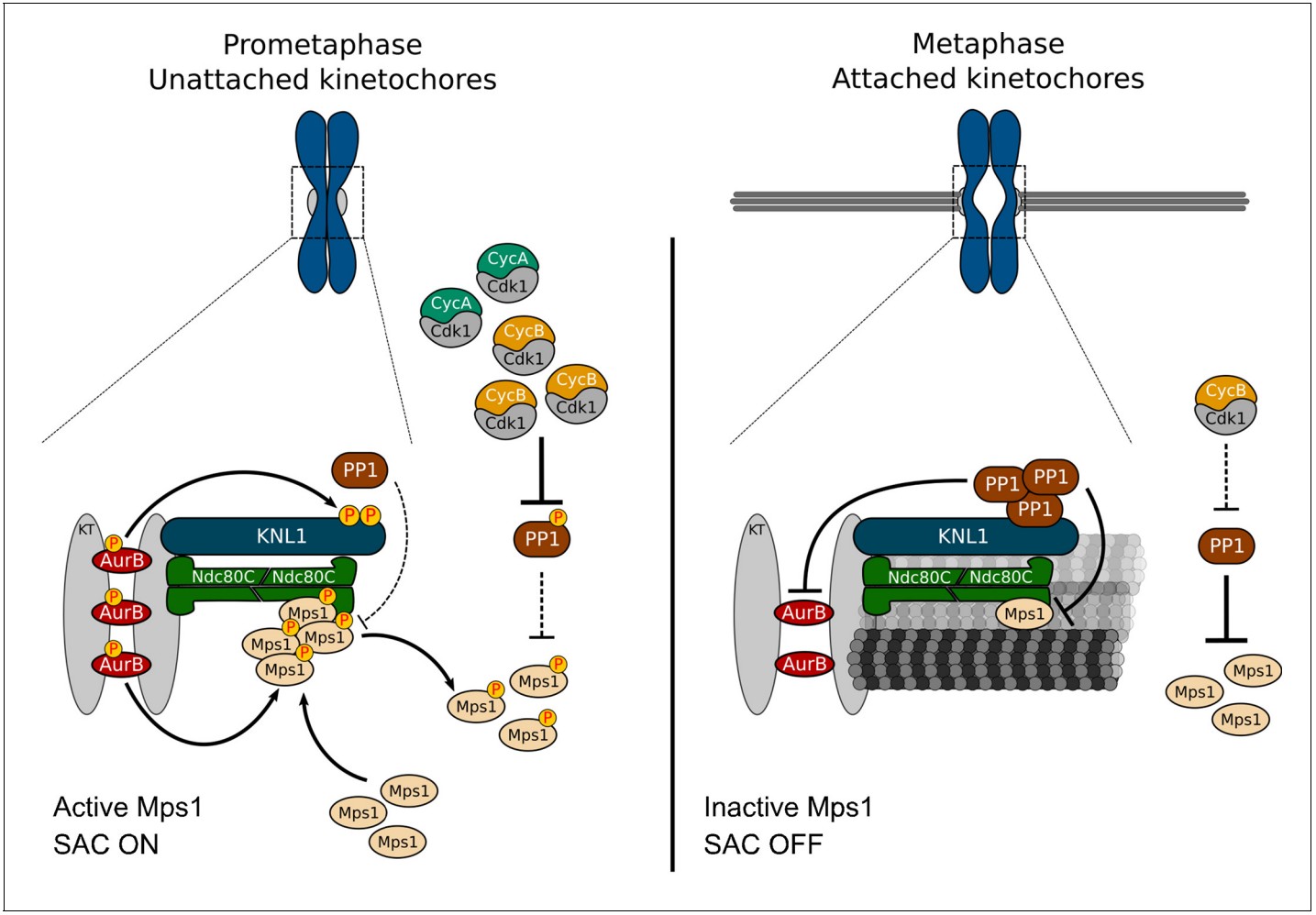

**Figure 7.** A model for regulation of Mps1 activation/inactivation in mitosis. Proposed model for regulation of soluble- and kinetochore-Mps1 activation in mitosis. During prometaphase, Aurora B potentiates the recruitment of Mps1 to unattached kinetochores and phosphorylates KNL1/Spc105 to limit PP1 kinetochore association. This allows the accumulation of active Mps1 at unattached kinetochores to instate efficient MCC assembly. In parallel, high levels of active CDK1 repress PP1 activity in the cytoplasm, which prevents the dephosphorylation and consequently inactivation of soluble Mps1. Active Mps1 in the cytoplasm promotes the assembly and/or prevents the disassembly of APC/C inhibitory complexes through mechanisms that are yet to be described. Cytoplasmic-and kinetochore generated MCC orchestrated by Mps1 cooperate to ensure efficient APC/C inhibition during unperturbed early mitosis. End-on attachment of microtubules mediated by the Ndc80 complex and KNL1 prevent the recruitment of Mps1 and exert tension across kinetochores and/or impose alterations on kinetochore architecture. Under these conditions, phosphorylation of KNL1 by Aurora B is minimal, and PP1 becomes enriched at bioriented kinetochores, where it dephosphorylates and inactivates the remaining residual pool of Mps1. Declining levels of active CDK1 allow PP1 to auto-activate and repress Mps1 activity in the cytoplasm, hence ensuring efficient SAC silencing and prompt anaphase onset.

EGFP-Mps1$^{WT}$ as template. PCR products were digested with DpnI restriction enzyme (New England Biolabs), used to transform competent bacteria and selected for positives. Mad1-EGFP construct under regulation of Mad1 native promoter cloned in the pMT backbone (Invitrogen) was a gift from Thomas Maresca (University of Massachusetts Amherst). The constructs H2B-GFP, mCherry-α-Tubulin and EGFP-Cyclin B have been previously described (*Conde et al., 2013*). Stable cell lines expressing the indicated constructs were obtained by cotransfection with the pCoBlast and selection in medium with 20 μg/ml blasticidin. Transfections of recombinant plasmids and pCoBlast into S2 cells were performed using Effectene Transfection Reagent (Qiagen), according to the manufacturer's instructions.

## Recombinant proteins

To generate pFastBac-Mps1$^{WT/KD}$ for expression of *Drosophila* Mps1$^{WT}$ (Wild type) and Mps1$^{KD}$ (Kinase-dead, D478A) in the baculovirus/Sf21 system, pMT-EGFP-Mps1$^{WT/KD}$ (*Conde et al., 2013*) and pFastBac HT A were digested with EcoRI and XbaI restriction enzymes (New England Biolabs). A 1414 bp fragment harboring Mps1$^{WT/KD}$ coding region was cloned into the destination vector. The obtained recombinant pFastBac-Mps$^{WT/KD}$ was used to transform DH10EMBacY competent cells in order to generate recombinant bacmid His6-Mps1$^{WT/KD}$. Positive clones were identified by blue/white screening and baculoviral DNA was isolated with QIAprep Spin Miniprep Kit (QIAGEN, Hilden, Germany) and used to transfect $10^6$ Sf21 cells in 6-well plate. Forty-eight hours after transfection, the supernatant was collected ($V_0$) and used to infect Sf21cells. Cell cultures in shaker flasks were split every 24 hr until proliferation arrest. Protein expression was monitored by YFP fluorescence and confirmed by SDS-PAGE. Cells were harvested 48 hr post infection and lysed in 50 mM Tris-HCl pH 7.5, 500 mM NaCl, 10 mM Imidazole. Lysates were sonicated and clarified by centrifugation at 4°C. Recombinant His6-Mps1$^{WT/KD}$ was purified by affinity chromatography in a His-Trap$^{TM}$-HP nickel-column (Amersham, Little Chalfon, United-Kingdom). The protein was eluted with 250 mM Imidazole and dialysed against 20 mM Hepes pH 7.5, 150 mM NaCl, 10 mM MgCl2, 1 mM DTT. To generate MBP-PP1-87B constructs for expression in bacteria, PCR products harboring PP1-87B coding sequence were cloned into StuI/XbaI sites of pMal-c2 (New England Biolabs) vector. Recombinant constructs were used to transform BL21-star competent cells and protein expression induced with 0.1 mM IPTG at 15°C, overnight. Lysates were sonicated and clarified by centrifugation at 4°C. Recombinant MBP-PP1-87B was purified with amylose magnetic beads (New England Biolabs) and eluted with 10 mM Maltose, 200 mM NaCl, 20 mM Tris-HCl pH 7.4, 1 mM EDTA and 1 mM DTT. To generate 6xHis-N-Mps1$^{WT}$ and 6xHis-N-Mps1$^{K231A/F234A}$ constructs for expression in bacteria, PCR products with the coding sequence corresponding to N-terminus 104–330 amino acids from Mps1$^{WT}$ and Mps1$^{K231A/F234A}$ were cloned into KpnI/XhoI sites of pET30a (+) vector (Novagen, Darmstadt, Germany). Recombinant constructs were used to transform BL21-star competent cells and protein expression induced with 0.05 mM IPTG at 15°C, overnight. Lysates were sonicated and clarified by centrifugation at 4°C. Recombinant 6xHis-N-Mps1$^{WT}$ and 6xHis-N-Mps1$^{K231A/F234A}$ were purified with Novex Dynabeads (Invitrogen) and eluted with 50 mM Imidazole, 200 mM NaCl, 20 mM Tris-HCl pH 7.4, 1 mM EDTA and 1 mM DTT. Recombinant human Mps1/TTK, PP1-γ and λ-Phosphatase were purchased from SignalChem, MRC-PPU Reagents (University of Dundee) and New England Biolabs, respectively.

## S2 cell lysates, pull-downs, immunoprecipitation and western blotting

For preparation of S2 cell lysates used in pull-down and immunoprecipitation experiments, S2 cultured cells overexpressing EGFP-Mps1$^{WT}$ and EGFP-Mps1$^{K231A/F234A}$ were treated with colchicine for 12 hr. Overexpression of EGFP-Mps1$^{WT}$ and EGFP-Mps1$^{K231A/F234A}$ was induced with 0.1 mM CuSO4 during 12 hr. Cells were harvested through centrifugation at 5000 rpm for 10 min at 4°C and afterwards washed with 2 mL PBS supplemented with protease inhibitors cocktail (Roche, Basel, Switzerland). Cell pellet was resuspended in lysis buffer (150 mM KCl, 75 mM HEPES, pH 7.5, 1.5 mM EGTA, 1.5 mM MgCl2, 15% glycerol, 0.1% NP-40, 1× protease inhibitors cocktail (Roche) and 1× phosphatase inhibitors cocktail 3 (Sigma)) before disruption through freezing in liquid nitrogen. Cell lysates were then clarified through centrifugation at 8000 rpm for 10 min at 4°C and quantified by Bradford protein assay (Bio-Rad). EGFP-Mps1$^{WT}$ and EGFP-Mps1$^{K231A/F234A}$ were immunoprecipitated from 0.5 mg of total cell lysates using the GFP-Trap_MA system (Chromotec GmbH, Planegg-Martinsried, Germany) according to the manufacturer's instructions. For pull-down assays using S2 cell protein extracts, 0.5 mg of total cell lysates were diluted in a final volume of 500 µl of column buffer (250 mM NaCl, 20 mM Tris-HCl pH 7.4, 1 mM EDTA, 1 mM DTT, 0.05% Tween-20, 1× protease inhibitors cocktail (Roche) and 1× phosphatase inhibitors cocktail 3 (Sigma-Aldrich)) and incubated with 5 µl of MBP or MBP-PP1-87B bound to amylose magnetic beads (New England Biolabs) for 90 min at room temperature with rotation. The magnetic beads and bound protein fraction were collected and washed 3 times with 750 µl of column buffer. For pull-down assays with purified recombinant proteins, 5 µl of 6xHis-N-Mps1$^{WT}$ or 6xHis-N-Mps1$^{K231AF234A}$ were incubated with 3 µl of MBP or MBP-PP1-87B bound to amylose magnetic beads (New England Biolabs) in a final volume of 30 µl of column buffer (250 mM NaCl, 20 mM Tris-HCl pH 7.4, 1 mM EDTA, 1 mM DTT) for

60 min at room temperature with rotation. The magnetic beads and bound protein fraction were collected and washed 3 times with 100 µl of column buffer. Magnetic beads and bound protein were resuspended in Laemmli sample buffer and boiled for 5 min at 95°C. After removal of the magnetic beads, samples were resolved by SDS-PAGE and probed for proteins of interest through western blotting. For western blot analysis, resolved proteins were transferred to a nitrocellulose membrane, using the iBlot Dry Blotting System (Invitrogen) according to the manufacturer's instructions. Transferred proteins were confirmed by Ponceau staining (0.25% Ponceau S in 40% methanol and 15% acetic acid). The membrane was blocked for 3 hr at room temperature with 5% dry milk in PBS-T. All the primary and secondary antibodies were diluted in PBS-T containing 1% dry milk and the membranes were incubated overnight at 4°C under agitation, then washed three times 10 min with PBS-T and immediately incubated with secondary antibodies for 1 hr at room temperature under agitation. Secondary antibodies conjugated to Horseradish peroxidase (Amersham) were used according to the manufacturer´s instructions. Blots were developed with ECL Chemiluminescent Detection System (Amersham) according to manufacturer's protocol and detected on X-ray film (Fuji Medical X-Ray Film).

### *In vitro* kinase and phosphatase assays

For *in vitro* kinase assays, 6xHis-DmMps1$^{WT}$, 6xHis-DmMps1$^{KD}$, HsMps1/TTK (0.25 µg) (SignalChem, Richmond, Canada) or immunoprecipitated EGFP-Mps1 were incubated for 30 min in a total volume of 15–30 µL kinase reaction buffer (5 mM MOPS, pH 7.2, 2.5 mM $\beta$-glycerol-phosphate, 5 mM MgCl2, 1 mM EGTA, 0.4 mM EDTA, 0.25 mM DTT, 100 µM ATP. Reactions were carried out at 25°C for His6-DmMps1$^{WT}$, His6-DmMps1$^{KD}$ and immunoprecipitated EGFP-Mps1 and at 30°C when using HsMps1/TTK. For inhibition of HsMps1/TTK activity, AZ3146 (Tocris, Bristol, United-Kingdom) was added to the reaction at a final concentration of 10 µM. For phosphatase assays, recombinant human PP1-$\gamma$ was added either to the kinase reaction at t0 or after Mps1/TTK inhibition. Reactions were carried out at 30°C for 30 min in PMP phosphatase buffer (New England Biolabs) containing 50 mM HEPES pH 7.5, 100 mM NaCl, 2 mM DTT, 0.01% Brij 35, 1 mM MnCl2. Reactions were stopped by addition of Laemmli sample buffer (4% SDS, 10% mercaptoetanol, 0.125 M Tris-HCl, 20% glycerol, 0.004% bromophenol blue) and heated for 5 min at 95°C. Peptides were resolved in an 8% SDS-PAGE. The activating autophosphorylation on T676 or T490 (for human Mps1/TTK and Drosophila Mps1, respectively) were detected by western blotting with a phospho-specific antibody provided by Geert Kops (*Jelluma et al., 2008*).

### Immunofluorescence analysis

For immunofluorescence analysis of S2 cells, $10^5$ cells were centrifuged onto slides for 5 min, at 1500 rpm (Cytospin 2, Shandon), and simultaneously fixed and extracted in 3.7% formaldehyde (Sigma-Aldrich), 0.5% Triton X-100 in PBS for 10 min followed by thee washing steps in PBS-T (PBS with 0.05% Tween 20) for 5 min each. Immunostaining was performed as described previously (*Conde et al., 2013*). Images were collected in Leica TCS SP5 II laser scanning confocal microscope (Leica Microsystems, Germany). Data stacks were analyzed and projected using ImageJ software (http://rsb.info.nih.gov/ij/). For immunofluorescence quantification, the mean pixel intensity was obtained from maximum projected raw images acquired with fixed exposure acquisition settings. For kinetochore proteins, the mean fluorescence intensity was quantified for individual kinetochores, selected manually by CID or Spc105 staining. The size of the region of interest (ROI) was predefined so that each single kinetochore could fit into. After subtraction of background intensities, estimated from regions of the cell with no kinetochores, the intensity was determined relative to CID or Spc105 reference and averaged over multiple kinetochores.

### Live cell imaging

Live analysis of mitosis was done in S2 cell lines expressing the indicated constructs. 4D datasets were collected at 25°C with a spinning disc confocal system (Revolution; Andor) equipped with an electron multiplying charge-coupled device camera (iXonEM+; Andor) and a CSU-22 unit (Yokogawa) based on an inverted microscope (IX81; Olympus). Two laser lines (488 and 561 nm) were used for near-simultaneous excitation of EGFP and mCherry. The system was driven by iQ software (Andor). Time-lapse imaging of z stacks with 0.8 µm steps covering the entire volume of the cell

were collected and image sequence analysis and video assembly done with ImageJ and iQ software. For quantification of EGFP-Mad1 fluorescence at kinetochores, the mean intensity was calculated within the area corresponding to kinetochores and corrected for cytosolic signal. The changes in fluorescence intensity with time were plotted as normalized signal relative to the signal measured at NEB.

## Analysis of microtubule dynamics by inducible fluorescent speckle microscopy

Inducible speckle imaging (ISI) was used to measure spindle microtubule turnover in *Drosophila* S2 cells as previously described (*Pereira et al., 2016*). Speckle patterns were imprinted using ~200- to 500-ms-long pulses with a bleaching strength (γ) between 1.5 and 2, a regime which maximizes speckle fluorescence amplitude. A rectangular ROI was defined enclosing a significant portion of the spindle area. Intensity contrast was then determined at each time point of the acquired sequence after the subtraction of a mean dark reference level from each frame. The microtubule turnover half-time was calculated by fitting a double-exponential curve ($y0 +A1*exp[-(t - t0)/\tau1]+A2 *exp[-(t - t0)/\tau2]$) to the mean intensity squared-contrast from independent cells for each time point. Before averaging, contrast curves were normalized to a 0-to-1 contrast change defining the pre- and post-ISI transition.

## FRAP experiments

FRAP experiments were performed in S2 cells expressing EGFP-MPS1[WT], EGFP-MPS1[KD] or EGFP-MPS1[K231A/F234A]. When required, colchicine was added to cultures at least 2 hr prior FRAP experiment. Datasets were collected at 25°C with a Leica TCS SP5II scanning confocal microscope (Leica Microsystems, Germany). The EGFP tag was excited using the 488 nm laser line set to 10% and bleached with the 405 nm laser line. Single kinetochores were bleached for three iterations once the EGFP fluorescence signal had become stable. Fluorescence intensity in the bleached area was acquired every 537 ms, 555 ms or 777 ms before and after bleaching. FRAP data analysis was performed with ImageJ. For each measurement the average fluorescence intensity in the bleached area was corrected for background and the ratio to average fluorescence in the cytoplasm determined. Average values before bleaching were set to 100%. The exponential kinetics of FRAP were analyzed by calculating the normalized unrecovered fluorescence at each time point (Finf-F(t))/(Finf-F(0)) where Finf is the value reached at the plateau, F(0) is the value observed in the first frame after bleaching and F(t) is the value at a given time point. FRAP kinetics parameters were determined by one phase exponential association curve fitting to normalized data using GraphPad Prism software.

## Fly stocks

All fly stocks were obtained from Bloomington Stock Center (Indiana, USA), unless stated otherwise. The *mps1* mutant allele *ald*[G4422] has been described before (*Conde et al., 2013*). *insc-GAL4* was used to drive the expression of *UAS-PP1-87B*[RNAi] in neuroblasts from brains of 3rd instar larvae brains. *w1118* was used as wild-type control. Fly stocks harboring *gEGFP-MPS1*[WT] and *gEGFP-MPS1*[KD] (D478A) under control of *Mps1 cis*-regulatory region were kindly provided by Christian Lehner (*Althoff et al., 2012*).

## Antibodies

The following primary antibodies were used for immunofluorescence studies: mouse anti-α-tubulin B512 (Sigma-Aldrich, RRID:AB_86546) used at 1:4000; rat anti-CID (Rat4) used at 1:250; guinea pig anti-Mps1 (Gp15) (a gift from Scott Hawley, RRID:AB_2567774) used at 1:250; rabbit anti-phosphorylated ser10-Histone H3 (p-H3) (Milipore, Billerica, MA, RRID:AB_565299) used at 1:5000; rabbit anti-Mad1 (Rb1, *Conde et al., 2013*) used at 1:2500; rabbit anti-phosphorylated Thr676-Mps (T676) (a gift from Geert Kops) used at 1:10000; rabbit anti-phosphorylated Thr232-Aurora B (Rockland, Limerik, PA) used at 1:1000; sheep anti PP1-87B (MRC-PPU Reagents, University of Dundee) used at 1:5000; rabbit anti-BubR1 (Rb 666, RRID:AB_2566989) used at 1:3000; rabbit anti-CenpC (Rb1) used at 1:5000; rat anti-Spc105 used at 1:250 and rabbit anti-Ndc80 (Rb272) (a gift from Michael Goldberg) used a 1:5000. The following primary antibodies were used for western blotting studies: mouse anti-α-tubulin DM1A (Sigma-Aldrich, RRID:AB_477593) used at 1:10000; mouse anti-Mps1

(NT clone 3-472-1, Milipore, RRID:AB_309903) used at 1:5000; guinea pig anti-Mps1 (Gp15) (a gift from Scott Hawley, RRID:AB_2567774) used at 1:5000; rabbit anti-phosphorylated Thr676-Mps1 (T676) (a gift from Geert Kops) used at 1:2000; mouse anti-MBP (New England Biolabs, RRID:AB_1559738) used at 1:5000; rabbit anti-BubR1 (Rb 666, RRID:AB_2566989) used at 1:3500; rabbit anti-CenpC (Rb1) used at 1:8000; mouse anti-GST (B-14) (Santa Cruz Biotech, Santa Cryz, CA, RRID:AB_627677) used at 1:2000.

### Statistical and kinetics analysis

All statistical and enzyme kinetics analysis was performed with GraphPad Prism V6.0f (Graph- Pad Software, Inc.).

### Supplementary information

The phospho-antibody that recognizes human Mps1 T676 autophosphorylation can be used to detect the conserved activating T-loop autophosphorylation on Drosophila Mps1 (T490). The phospho-antibody generated against human Mps1 T676 autophosphorylation readily detected in immunoblots the autophosphorylation of a recombinant wild type version of *Drosophila* Mps1 (His6-Mps1$^{WT}$) contrasting with the absence of evident phosphorylation on a catalytically inactive version of the kinase (His6-Mps1$^{KD}$) (*Figure 2—figure supplement 2B*). Moreover, immunofluorescence analyses of 3$^{rd}$ instar larval neuroblasts revealed a significant decrease in Mps1 T490 phosphorylation at kinetochores of *ald*$^{G4422}$ strong hypomorphic *Mps1* mutants (*Figure 2—figure supplement 2C, D*). Importantly, a similar reduction was observed following the expression of kinase dead EGFP-Mps1 in *ald*$^{G4422}$ background when compared with mutants expressing the Mps1 wild-type transgene (*Figure 2—figure supplement 2E,F*). Taken together, these data validate the phospho-antibody as a read-out for the activation status of *Drosophila* Mps1.

## Acknowledgements

We thank Geert Kops (UMC, Utrecht, The Netherlands), Thomas Maresca (University of Massachusetts Amherst, USA) and Christian Lehner (University of Zurich, Switzerland) for antibodies, constructs and fly stocks. We thank Carla Sofia Lopes (i3S, IBMC, University of Porto) and all the members of the Sunkel laboratory for critical and helpful discussions. This article is a result of the project Norte-01-0145-FEDER-000029 - Advancing Cancer Research: From basic knowledge to application, supported by Norte Portugal Regional Operational Programme (NORTE 2020), under the PORTUGAL 2020 Partnership Agreement, through the European Regional Development Fund (FEDER). MO is supported by a fellowship from the GABBA PhD program from the University of Porto, PD/BD/105746/2014. JB is supported by an FCT PhD grant SFRH/BD/87871/2012. CC is supported by an FCT investigator position and funding (IF/01755/2014). HM is funded by PRECISE and CODECHECK grants from the European Research Council, FLAD Life Science 2020, and the Louis-Jeantet Young Investigator Career Award.

## Additional information

### Funding

| Funder | Grant reference number | Author |
|---|---|---|
| FEDER-Fundo Europeu de Desenvolvimento Regional funds through the COMPETE 2020 | Norte-01-0145-FEDER-000029 | Margarida Moura<br>Claudio E Sunkel<br>Carlos Conde |
| Fundação para a Ciência e a Tecnologia | PTDC7BEX-BCM/192½014-PR041602 | Claudio E Sunkel |
| Fundação para a Ciência e a Tecnologia | PEst-C/SAU/LA0002/2013-Incentivo2014-BGCT | Claudio E Sunkel |
| Fundação para a Ciência e a Tecnologia | FCT Investigator grant IF/01755/2014 | Carlos Conde |
| Fundação para a Ciência e a | GABBA PhD Program grant | Mariana Osswald |

| | | |
|---|---|---|
| Tecnologia | PD/BD/105746/2014 | |
| Fundação para a Ciência e a Tecnologia | FCT PhD grant SFRH/BD/ 87871⁄2012 | João Barbosa |
| European Research Council | PRECISE | Helder Maiato |
| European Research Council | CODECHECK | Helder Maiato |
| FLAD Life Science | | Helder Maiato |

The funders had no role in study design, data collection and interpretation, or the decision to submit the work for publication.

## Author contributions

MM, MO, Formal analysis, Investigation, Methodology; NL, JB, Investigation, Methodology; AJP, Resources, Methodology; HM, Resources, Validation, Writing—review and editing; CES, Conceptualization, Resources, Supervision, Funding acquisition, Validation, Writing—review and editing; CC, Conceptualization, Formal analysis, Supervision, Funding acquisition, Investigation, Methodology, Writing—original draft, Writing—review and editing

## Author ORCIDs

Carlos Conde, http://orcid.org/0000-0002-4177-8519

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
