## [Decision Letter]

Thank you for submitting your article "Protein Phosphatase 1 inactivates Mps1 to ensure efficient Spindle Assembly Checkpoint silencing" for consideration by *eLife*. Your article has been favorably evaluated by Anna Akhmanova (Senior Editor) and three reviewers, one of whom, Andrea Musacchio (Reviewer #1), is a member of our Board of Reviewing Editors.

The reviewers have discussed the reviews with one another and the Reviewing Editor has drafted this decision to help you prepare a revised submission.

Summary:

In this manuscript, Moura, Osswald et al. investigated an important unresolved problem in the spindle assembly checkpoint (SAC) field: how the wait-anaphase signal is inactivated in a timely manner once the SAC has been satisfied. Specifically, the manuscript describes a new facet of the complex regulation of the SAC kinase Mps1. The authors contend that depletion of the *Drosophila melanogaster* (dm) Protein Phosphatase 1 (PP1) ortholog PP1-87B leads to a mitotic arrest in metaphase, and that this correlates in turn with the presence of the active, phosphorylated form of Mps1 on the kinetochores of aligned metaphase chromosomes, for all apparent levels of tension (using inter-kinetochore distance as its surrogate marker). This is an important observation, because current models based on two 2015 papers in Science by the Kops and Yu laboratories have proposed that microtubule binding prevents kinetochore recruitment of Mps1 through direct competition. At face value, the authors' observations appear inconsistent with this previously proposed model. The authors also make the compelling point that *Drosophila* cells are the ideal model system to analyze this SAC regulatory mechanism because Dm Spc105/KNL1 MELT-like motifs are not subjected to phospho-regulation, yet PP1 is required for mitotic exit, pointing to Mps1 as a possible crucial target. Indeed, the authors focus on a possible physical interaction of PP1 with Mps1, which ultimately results in the dephosphorylation of the T-loop of Mps1 in metaphase, ensuring timely SAC inactivation. The authors contend that the expression of an Mps1 mutant that escapes this interaction largely phenocopies the depletion of PP1. In addition, the authors show that regulation by Mps1 is not only affecting the kinetochore pool of the kinase, but also extends to the cytosolic form. The authors suggest that this regulation may not be limited to *Drosophila* Mps1, and that it could be more general.

All three reviewers recognise the significant interest and potential importance of the work. However, they also agree that the data provided do not unequivocally show that Mps1 is a direct substrate of PP1. Main points of criticism are:

1) In *Drosophila* the PP1-87B isoform accounts for about three quarters of all PP1 activity. Hence, the depletion of PP1-87B will increase the phosphorylation of hundreds of substrates and affect nearly every cellular process. Some of the observed effects may even be mediated by other phosphatases (e.g. PP2A), which are known to be regulated by PP1. Therefore, an increased phosphorylation of Mps1 by the depletion of PP1-87B can at best be used as preliminary evidence for an (in)direct role of PP1 in the dephosphorylation and inactivation of Mps1.

2) One way to delineate the exact contribution of PP1 to Mps1 dephosphorylation is by interference with the involved PP1-targeting subunit. The authors conclude that PP1 directly interacts with Mps1 via a canonical RVxF motif. However, the data are not conclusive. First, the consensus binding motif for PP1 is not conserved in other organisms (Figure 3) and it is not clear whether this motif resides in an intrinsically disordered region of Mps1, which is a requirement for binding to PP1. Second, RVxF and SILK consensus sequences are present in a third of all proteins and their mere presence does not imply that they also function as PP1 binding motifs. In fact, most consensus PP1 binding sequences do not interact with PP1, often because they reside in structured domains. Most importantly, a PP1 binding mutant of Mps1 still binds nearly normal amounts of PP1 (Figure 3) and it is therefore not clear how to explain the effects of the expression of this Mps1 mutant. All three reviewers raised this point, which is therefore critical. At present, it cannot be concluded that the increased phosphorylation of Mps1 at its T-loop following mutation of the RVxF sequence is due to a loss of associated PP1. This also implies that the molecular mechanism underlying the metaphase delay effects after expression of the Mps1 K231A/F234A mutant remains unaccounted for.

Essential revisions:

1) All reviewers felt that the pulldowns in Figure 3 are not convincing. It definitely looks like the amount of MBP-PP187B is significantly higher in the WT extracts. The levels of EGFP to MBP were ratioed, but the data are not strongly convincing, as the EGFP looks quite comparable between the WT and mutant. The cell-based data on this mutant are more convincing than the results obtained *in vitro*. However, with the present data, one of the paper's main hypotheses appears unsubstantiated, and the reviewers recommend a more detailed characterisation of the Mps1-PP1 interaction.

2) The depletion of SDS22 mimicked PP1-87B RNAi (Figure 2—figure supplement 3). This interesting observation was not explored further. In view of the criticisms raised in point 1, it is possible that SDS22 mediates the binding of PP1-87B by Mps1. This possibility could be tested relatively easily.

3) The purified catalytic subunit of PP1 will dephosphorylate nearly every phosphoprotein when added in sufficient amounts, which makes the experiments shown in Figure 3 inconclusive. It would have been much more informative to compare the dephosphorylation of Mps1-WT and Mps1-K231A/F234A by limiting concentrations of PP1. If Mps1 directly recruits PP1 via its putative RVxF motif (but see point above) the WT protein can be expected to be dephosphorylated at lower PP1 concentrations than the K231A/F234A mutant.

4) Figure 3—figure supplement 1 is taken as evidence that the λ phosphatase does not dephosphorylate Mps1 at T676. However, panel C shows much faster dephosphorylation with λ phosphatase than with PP1. Likewise, in Figure 3, the major band disappears with λ phosphatase. The phosphatase units should be clearly defined: a comparison based upon units only makes sense if they are defined in the same manner.

5) A central finding is that PP1 depletion leads to delay in SAC satisfaction due to excess active p-Mps1 at kinetochores (and in the cytosol). Mad1 is also retained at these kinetochores. A possible explanation for this phenotype is that elevated aurora B kinase, as a consequence of PP1 depletion, simply generates unattached kinetochores that retain Mad1 and delay the cell in metaphase. Key in this regard is to convincingly demonstrate that fully stable attachments are formed. To strengthen this conclusion, the authors should better quantify the cold stability data. All that is shown for these experiments are a few images of cold-treated cells.

6) Related to point 5, turnover measurements after photo-activation have been used more commonly, and the authors should elaborate on the validity and robustness of the ISI protocol. Has the ISI approach been validated by comparison to PA measurements? In Figure 1, the slow turnover component corresponding to KT-MTs has a relatively short half-life in comparison to those reported in previous studies (extending to several minutes). What may be the reason for this? If PA measurements could be considered, they would strengthen this line of enquiry, which is critical for the study. A supplemental video of a representative ISI experiment would be helpful.

Furthermore, the authors may consider the following two comments:

7) The pMps1-T490P seems to localize to the inner centromere rather than the kinetochore in the *w1118* larval neuroblasts and the EGFP-WT Mps1 in ald/ald neuroblasts. A CID internal (inner centromere?) Mps1-T490P staining pattern is also observed in Figure 6. Can the authors share their thoughts on the relevance of Mps1 localization patterns? For example, are there multiple Mps1 sub-populations: cytosolic, kinetochore, inner centromere?

8) Figure 6 should be discussed in the Results section, not in the Discussion. The authors speculate that the phosphorylation of PP1 by Cdk1 (at the C-terminus, not the N-terminus as indicated in the fourth paragraph of the Discussion) contributes to keeping Mps1 active at the beginning of mitosis. However, there are no data supporting the notion that Mps1-associated PP1-87B is actually phosphorylated by Cdk1 at the beginning of mitosis. More generally, regulation of PP1 by Cdk1 is unlikely to contribute to the switch from the SAC on to the SAC off condition, as the SAC is switched off before Cyclin B destruction can start, i.e. it is switched off under conditions of high Cdk1 activity. Furthermore, while the focus is on CDK1/CyclinB-mediated regulation of PP1, CDK1/CyclinA is also likely contributing to this pathway during prometaphase. Please add some discussion of the contributions of Cyclin A during prometaphase. Does MG132-treatment block the increase in kinetochore-associated PP1 at aligned kinetochores that is nicely shown in Figure 6?

---

## [Author Response]

Essential revisions:

*1) All reviewers felt that the pulldowns in Figure 3 are not convincing. It definitely looks like the amount of MBP-PP187B is significantly higher in the WT extracts. The levels of EGFP to MBP were ratioed, but the data are not strongly convincing, as the EGFP looks quite comparable between the WT and mutant. The cell-based data on this mutant are more convincing than the results obtained* in vitro*. However, with the present data, one of the paper's main hypotheses appears unsubstantiated, and the reviewers recommend a more detailed characterisation of the Mps1-PP1 interaction.*

We agree with the reviewers and following their recommendation we performed a more detailed characterization of the interaction between Mps1 and PP1. We repeated the pull-downs, but this time under more stringent conditions (pull-down buffer supplemented with 250 mM NaCl + Tween 0.05%). This allowed us to clearly show that the interaction between Mps1 and PP1-87B is severely compromised when the KVLF motif is mutated to AVLA (Mps1^K231A/F234A^). We have presented these data and its corresponding quantification in Figure 3 of the revised manuscript.

To demonstrate direct binding of Mps1 to PP1-87B and the requirement of the KVLF motif for such an association we produced recombinant fragments of Mps1 N-terminus (104-330 aa) harboring the KVLF motif and its mutated version AVLA. MBP-PP1-87B was able to pull-down the recombinant Mps1^WT^ N-terminus as opposed to the Mps1^K231A/F234A^ fragment. These results demonstrate a direct interaction between Mps1 N-terminus and PP1-87B that is mediated by a canonical RVxF motif. We have included these new data in Figure 3 of the revised manuscript.

2) The depletion of SDS22 mimicked PP1-87B RNAi (Figure 2—figure supplement 3). This interesting observation was not explored further. In view of the criticisms raised in point 1, it is possible that SDS22 mediates the binding of PP1-87B by Mps1. This possibility could be tested relatively easily.

We agree that the increase of Mps1 T-loop phosphorylation upon Sds22 depletion is an interesting observation. Following the reviewers’ suggestion, we assessed whether Sds22 might be necessary for the interaction between Mps1 and PP1-87B. For that, we performed pull-downs using MBP-PP1-87B and lysates from S2 cells expressing EGFP-Mps1^WT^ and depleted of Sds22. Binding of EGFP-Mps1^WT^ to MBP-PP1-87B is readily detected when lysates of control cells are used. Notably, the absence of Sds22 decreased this interaction. These results are consistent with a role for Sds22 in mediating the interaction between PP1-87B and full-length Mps1 and have been included in Figure 3 of the revised manuscript. Thus, although Mps1 N-terminus is able to bind PP1-87B in vitro through a canonical RVxF motif (please see response to point 1), the interaction of PP1-87B with full-length Mps1 from cell lysates seems to require the involvement of SdS22 regulatory subunit.

3) The purified catalytic subunit of PP1 will dephosphorylate nearly every phosphoprotein when added in sufficient amounts, which makes the experiments shown in Figure 3 inconclusive. It would have been much more informative to compare the dephosphorylation of Mps1-WT and Mps1-K231A/F234A by limiting concentrations of PP1. If Mps1 directly recruits PP1 via its putative RVxF motif (but see point above) the WT protein can be expected to be dephosphorylated at lower PP1 concentrations than the K231A/F234A mutant.

We thank the reviewers for this important suggestion. To address this, we immunoprecipitated EGFP-Mps1^WT^ and EGFP-Mps1^K231A/F234A^ from lysates of S2 cells depleted of endogenous PP1-87B. Immunoprecipitated Mps1 was incubated with ATP and with increasing concentrations of PP1. We found that the T-loop of Mps1^WT^ was dephosphorylated at lower PP1 concentrations than the Mps1^K231A/F234A^ mutant, thus validating KVLF as a PP1-docking motif and Mps1 as a direct substrate of PP1. We have presented these data on Figure 3 of the revised manuscript.

4) Figure 3—figure supplement 1 is taken as evidence that the λ phosphatase does not dephosphorylate Mps1 at T676. However, panel C shows much faster dephosphorylation with λ phosphatase than with PP1. Likewise, in Figure 3, the major band disappears with λ phosphatase. The phosphatase units should be clearly defined: a comparison based upon units only makes sense if they are defined in the same manner.

The panel C on Figure 3—figure supplement 1 of the original manuscript does not depict the dephosphorylation of Mps1 T-loop but instead the dephosphorylation of an artificial substrate by λ phosphatase and by PP1. As described in the corresponding figure legend, the graph represented the dephosphorylation of DiFMUP over time in an EnzChek phosphatase assay. This control experiment was performed to demonstrate that although at the indicated units of λ phosphatase failed to efficiently dephosphorylate Mps1-T loop the phosphatase was nevertheless active towards other phospho-substrate. We apologize for not making it clear and to avoid misinterpretation we decided to remove these data from the revised version of the manuscript.

We agree with the reviewers that the comparison between PP1 and λ phosphatase based upon units only makes sense when these are defined in the same manner. Therefore, we repeated most of the experiments using PP1 and λ phosphatase at the same molar concentration, which is unambiguous and allows valid comparisons to be made. The results obtained confirmed a higher specificity of PP1 towards Mps1 T-loop and are presented in Figure 3 and Figure 3—figure supplement 1 of the revised manuscript.

5) A central finding is that PP1 depletion leads to delay in SAC satisfaction due to excess active p-Mps1 at kinetochores (and in the cytosol). Mad1 is also retained at these kinetochores. A possible explanation for this phenotype is that elevated aurora B kinase, as a consequence of PP1 depletion, simply generates unattached kinetochores that retain Mad1 and delay the cell in metaphase. Key in this regard is to convincingly demonstrate that fully stable attachments are formed. To strengthen this conclusion, the authors should better quantify the cold stability data. All that is shown for these experiments are a few images of cold-treated cells.

We agree with the reviewers. Following their suggestion, we quantified the% of kinetochores bound to cold-resistant microtubule bundles in control, PP1-87B- and Ndc80-depleted cells, as well as in cells expressing EGFP-Mps1^WT^, EGFP-Mps1^K231A/F234A^. The results from the quantifications are depicted in Figure 1 and Figure 4—figure supplement 1 of the revised manuscript and clearly demonstrate that fully stable attachments are formed upon depletion of PP1-87B or expression of Mps1^K231A/F234A^.

6) Related to point 5, turnover measurements after photo-activation have been used more commonly, and the authors should elaborate on the validity and robustness of the ISI protocol. Has the ISI approach been validated by comparison to PA measurements? In Figure 1, the slow turnover component corresponding to KT-MTs has a relatively short half-life in comparison to those reported in previous studies (extending to several minutes). What may be the reason for this? If PA measurements could be considered, they would strengthen this line of enquiry, which is critical for the study. A supplemental video of a representative ISI experiment would be helpful.

We used ISI to measure microtubule turnover rates because this method is a robust alternative to ROI based techniques, where marking of tubulin is typically done by imaging a mask onto the sample to trigger the photoswitch, generating a pattern confined to the objective’s depth of focus. Although the pattern is constrained to the focal plane, out-of-focus planes still experience a (homogeneous) photoswitch, precluding sequential z-stack imprinting to generate a 3D pattern. In ISI, the fluorescence pattern after the pulse is 3D, hence unrelated to the particular focal plane at the time of the imprinting and thus covers the whole spindle with measurable speckle spots. Thus, different z-layers can be chosen after acquisition. This contributes to operational robustness and operational simplicity when compared to conventional ROI-switch techniques. Not the least, bright speckles decay and dark speckles recovery are globally and mutually measured through a statistical measure (intensity contrast), insensitive to global, acquisition-related, bleaching.

For a detailed description of ISI please consider the article by Pereira and colleagues (2016) [PMID: 26783303], where ISI has also been used to measure spindle microtubule turnover dynamics in *Drosophila* S2 cells. The half-life value determined by Pereira et al. (2016) for the slow turnover component in metaphase was 21 seconds, which is very similar to the reference values obtained in previous studies using photobleaching: Buster et al. (2007) [PMID: 17553931] and Goshima et al. (2008) [PMID: 18443220] respectively reported half-life values of ~ 20 seconds and 30 seconds for the stable population of metaphase microtubules in S2 cells. Moreover, a fly stock available in the lab (UASp-alphaTub84B.tdEOS) allowed us to determine during the revision period of the manuscript the turnover rates of spindle microtubule in metaphase *Drosophila* neuroblasts by photoconversion (~ 40 seconds). Thus, the ISI measurements described in our manuscript (~ 40-45 seconds) and in Pereira et al. (2016) for kinetochore-microtubules of *Drosophila* metaphase cells are comparable to the values obtained when photobleaching or photoconversion approaches are used. Collectively, these data support the validity and robustness of the ISI protocol as a method to measure microtubule turnover rates. Following the reviewers’ recommendation, we have provided supplemental videos of representative ISI experiments in the revised version of the manuscript (Video 4 and Video 5).

Since we resorted to ISI to benefit from its abovementioned advantages over the conventional ROI-based methods and the values that we obtained are very similar to the reference values reported for S2 cells, performing PA measurements to confirm the stability of kinetochore-microtubule interactions as suggested by the reviewers would only cover the purpose of using the ISI technique. Note that the ISI data is also supported by interkinetochore measurements and cold-stability quantifications that show fully stable attachments in PP1-87B depleted cells.

Furthermore, the authors may consider the following two comments:

7) The pMps1-T490P seems to localize to the inner centromere rather than the kinetochore in the w1118 larval neuroblasts and the EGFP-WT Mps1 in ald/ald neuroblasts. A CID internal (inner centromere?) Mps1-T490P staining pattern is also observed in Figure 6. Can the authors share their thoughts on the relevance of Mps1 localization patterns? For example, are there multiple Mps1 sub-populations: cytosolic, kinetochore, inner centromere?

We do not have an explanation for the inner-centromeric localization of Mps1-T490Ph observed in larval neuroblasts. One can speculate on the existence of an inner centromeric subpopulation of active Mps1 that is more evident in fly neuroblasts. However, at this point, we do not have solid data to confirm this notion. In S2 cells however, the phospho-antibody consistently stains the kinetochore. The inset on the bottom panel of Figure 6 of the original manuscript is misrepresenting the actual localization pattern. The inset depicts the magnification of two kinetochores from two different adjacent kinetochore pairs, thus wrongly suggesting a CID internal Mps1-T490Ph signal. We apologize for this and selected an appropriate pair of kinetochores from the same image, which is now presented on Figure 6 of the revised manuscript.

8) Figure 6 should be discussed in the Results section, not in the Discussion. The authors speculate that the phosphorylation of PP1 by Cdk1 (at the C-terminus, not the N-terminus as indicated in the fourth paragraph of the Discussion) contributes to keeping Mps1 active at the beginning of mitosis. However, there are no data supporting the notion that Mps1-associated PP1-87B is actually phosphorylated by Cdk1 at the beginning of mitosis. More generally, regulation of PP1 by Cdk1 is unlikely to contribute to the switch from the SAC on to the SAC off condition, as the SAC is switched off before Cyclin B destruction can start, i.e. it is switched off under conditions of high Cdk1 activity. Furthermore, while the focus is on CDK1/CyclinB-mediated regulation of PP1, CDK1/CyclinA is also likely contributing to this pathway during prometaphase. Please add some discussion of the contributions of Cyclin A during prometaphase. Does MG132-treatment block the increase in kinetochore-associated PP1 at aligned kinetochores that is nicely shown in Figure 6?

Following the reviewers’ suggestion we moved the discussion of the data presented in Figure 6 of the original manuscript to the Results section under the title “Regulation of PP1 activity during mitosis”. We have also added new data supporting that Aurora B activity limits the accumulation of PP1-87B at unattached/unaligned kinetochores. This is presented in Figure 6 of the revised manuscript. Furthermore, to address the reviewers’ concerns, we have provided a more critical discussion about the regulation of PP1 by CDK1/Cyclin B and of its relevance for SAC silencing. A possible contribution of CDK1/Cyclin A in controlling the PP1-mediated transition from the SAC on to the SAC off condition has been added to the Discussion.